# Multiphase Oxidation of $SO_2$ by $NO_2$ on $CaCO_3$ Particles

Defeng Zhao[*], Xiaojuan Song[*], Tong Zhu, Zefeng Zhang, Yingjun Liu

BIC-ESAT and SKL-ESPC, College of Environmental Sciences and Engineering, Peking University, Beijing, 100871, China

*These authors contributed equally to this work.

Correspondence to: Tong Zhu (tzhu@pku.edu.cn)

**Abstract.** Heterogeneous/multiphase oxidation of $SO_2$ by $NO_2$ on solid or aqueous particles is thought to be a potentially important source of sulfate in the atmosphere, for example, during heavily polluted episodes (haze), but the reaction mechanism and rate are uncertain. In this study, in order to assess the importance of the direct oxidation of $SO_2$ by $NO_2$ we investigated the heterogeneous/multiphase reaction of $SO_2$ with $NO_2$ on individual $CaCO_3$ particles in $N_2$ using Micro-Raman spectroscopy. In the $SO_2/NO_2/H_2O/N_2$ gas mixture, the $CaCO_3$ solid particle was first converted to the $Ca(NO_3)_2$ droplet by the reaction with $NO_2$ and the deliquescence of $Ca(NO_3)_2$, and then $NO_2$ oxidized $SO_2$ in the $Ca(NO_3)_2$ droplet forming $CaSO_4$, which appeared as needle-shaped crystals. Sulfate was mainly formed after the complete conversion of $CaCO_3$ to $Ca(NO_3)_2$, that is, during the multiphase oxidation of $SO_2$ by $NO_2$. The precipitation of $CaSO_4$ from the droplet solution promoted sulfate formation. The reactive uptake coefficient of $SO_2$ for sulfate formation is on the order of $10^{-8}$, and RH enhanced the uptake coefficient. We estimate that the direct multiphase oxidation of $SO_2$ by $NO_2$ is not an important source of sulfate in the ambient atmosphere compared with the $SO_2$ oxidation by OH in the gas phase and is not as important as other aqueous phase pathways, such as the reactions of $SO_2$ with $H_2O_2$, $O_3$, and $O_2$, with or without transition metals.

# 1 Introduction

Sulfate is a major component of atmospheric particulate matter. It contributes to a large fraction of atmospheric aerosol particles in both urban and rural areas (Seinfeld and Pandis, 2006; Zhang et al., 2007). Sulfate is either from primary source, such as sea spray, or from secondary source, i.e., by the oxidation of reduced sulfur compounds such as dimethyl sulfide (DMS), carbonyl sulfur (COS), and $SO_2$ (Seinfeld and Pandis, 2006). In the continent, the main source of sulfate is the oxidation of $SO_2$, an important air pollutant from fossil fuel combustion. $SO_2$ can be oxidized in the gas phase, mainly by OH, or in the particle phase such as by $H_2O_2$, $O_3$, or $O_2$ catalyzed by transition metal ions in cloud or fog water (Seinfeld and Pandis, 2006; Finlayson-Pitts and Pitts Jr., 1999) or by $O_3$ or photochemical reactions on particle surface (Zhu et al., 2011; Li et al., 2006; Li et al., 2007; Shang et al., 2010; Li et al., 2011).

Although various pathways of $SO_2$ oxidation are identified, the source of sulfate and relative importance of various pathways of $SO_2$ oxidation forming sulfate in the atmosphere still remain uncertain. For example, during heavily polluted episodes (haze) in China in recent years, high concentrations of sulfate were observed, but the source of sulfate is elusive (Wang et al., 2016; Wang et al., 2014a; Zheng et al., 2015b; Guo et al., 2014). The relative contribution of regional transport versus local formation and physical and chemical mechanisms responsible for sulfate formation are still not clear. Recent studies have highlighted heterogeneous reactions of $SO_2$ on solid or liquid particles to be a possibly important source of sulfate based on model, field and laboratory studies (Huang et al., 2014; Zhu et al., 2011; Gao et al., 2016; Zheng et al., 2015a; Wang et al., 2014b; He et al., 2014; Fu et al., 2016; Xue et al., 2016; Xie et al., 2015; Cheng et al., 2016; Wang et al., 2016). During haze episodes, relative humidity (RH) is often high (Zhang et al., 2014; Wang et al., 2016; Zheng et al., 2015b) and particles or some components of particles can deliquesce forming aqueous solution. In particular, several recent studies propose that the multiphase oxidation of $SO_2$ by $NO_2$, another important air pollutant, on liquid particles may be a major pathway of sulfate formation (Wang et al., 2016; Xue et al., 2016; Xie et al., 2015; Cheng et al., 2016). Both $SO_2$ and $NO_2$ are from fossil fuel combustion and both concentrations are often high during haze episodes, and their reaction may significantly contribute to sulfate formation.

In order to assess and quantify the role of the heterogeneous reactions of $SO_2$ in sulfate formation, laboratory studies are needed to understand the reaction process and obtain kinetic parameters for modeling such as uptake coefficients of $SO_2$. Among many studies investigating the heterogeneous reactions of $SO_2$ on various particles (Goodman et al., 2001; Li et al., 2011; Shang et al., 2010; Huang et al., 2015; Huang et al., 2016; Zhou et al., 2014; Li et al., 2004; Kong et al., 2014; Passananti et al., 2016; Cui et al., 2008; Chu et al., 2016; Zhao et al., 2015; Li et al., 2006; Wu et al., 2011; He et al., 2014; Liu et al., 2012; Ma et al., 2008; Park and Jang, 2016; Ullerstam et al., 2002; Sorimachi et al., 2001; Ullerstam et al., 2003; Wu et al., 2013; Wu et al., 2015), only a few have investigated the heterogeneous reaction of $SO_2$ in the presence of $NO_2$ (He et al., 2014; Liu et al., 2012; Ma et al., 2008; Park and Jang, 2016; Ullerstam et al., 2003; Ma et al., 2017). These studies found that $NO_2$ can promote sulfate formation from $SO_2$ oxidation (He et al., 2014; Liu et al., 2012; Ma et al., 2008; Park and Jang, 2016; Ullerstam et al., 2003). However, the mechanism of this effect is still not clear and only few studies reported kinetic parameters such as uptake coefficient of $SO_2$ due to the reaction with $NO_2$. Importantly, most of these studies focused on the gas-solid reactions on particles. Very few laboratory studies have investigated the multiphase reaction of $SO_2$ with $NO_2$ on atmospheric aqueous particles or solid-aqueous mixed phase aerosol

particles, and the uptake coefficient of $SO_2$ on atmospheric aqueous particles due to the reaction with $NO_2$ is largely unknown. From several decades ago until now, a number of studies have investigated the aqueous reaction of soluble S(IV) species ($H_2SO_3$, $HSO_3^-$, $SO_3^{2-}$) with $NO_2$ in dilute bulk solution (Lee and Schwartz, 1983; Clifton et al., 1988; Littlejohn et al., 1993; Takeuchi et al., 1977; Nash, 1979; Ellison and Eckert, 1984; Shen and Rochelle, 1998; Tursic and Grgic, 2001) relevant to the conditions in cloud water. However, in aqueous aerosol particles, the reaction rate and process may be substantially different from those in bulk solution due to high ionic strength resulted from high concentrations of solutes, potential interactions of sulfate with other ions, and low water activity in aerosol particles.

In this study, we present the finding that the multiphase reaction of $SO_2$ directly with $NO_2$ is not an important source of sulfate in the atmosphere, in the absence of other oxidants such as $O_2$. The direct oxidation of $SO_2$ by $NO_2$ pathway was proposed in a number of recent studies to be potentially important for sulfate formation (Cheng et al., 2016; Wang et al., 2016; Xue et al., 2016). For example, Cheng et al. (2016) considered the direct oxidation of $SO_2$ by $NO_2$ to be the most important pathway to explain the missing sulfate source during the haze events in Beijing. Wang et al. (2016) also proposed that the direct oxidation of $SO_2$ by $NO_2$ is key to efficient sulfate formation in the presence of high relative humidity and $NH_3$ and showed that in their laboratory study sulfate formation is mainly contributed by the direct oxidation by $NO_2$ and the role of $O_2$ is negligible.

We investigated the heterogeneous reaction of $SO_2$ with $NO_2$ on $CaCO_3$ particles at the ambient RH. $CaCO_3$ is an important component of mineral aerosols, especially in East Asia (Cao et al., 2005; Song et al., 2005; Okada et al., 2005) and it is a very reactive component (Krueger et al., 2004; Li et al., 2010; Li et al., 2006; Prince et al., 2007a). It is also one of the few alkaline particles in the atmosphere, especially in northern China, which can neutralize acids on particles and increase the pH of aerosol water, thus promoting the apparent solubility and uptake of $SO_2$. The reaction of $SO_2$ with $NO_2$ on $CaCO_3$ has been suggested by field observations, which showed internal mixing of $CaCO_3$, $CaSO_4$, and $Ca(NO_3)_2$ in particles (Hwang and Ro, 2006; Li and Shao, 2009; Zhang et al., 2000). More importantly, as shown below, during the reaction on $CaCO_3$, aqueous phase can be formed, which allows us to investigate the multiphase reaction of $SO_2$ with $NO_2$. We studied the reaction of $SO_2$ and $NO_2$ on individual $CaCO_3$ particles in $N_2$ using Micro-Raman spectrometer with a flow reaction system. $N_2$ was used as carrier gas in order to avoid confounding effects of other oxidants including $O_2$ in $SO_2$ oxidation. Combining the chemical and optical information from Micro-Raman spectrometer, we systematically investigated the reaction process and quantified the reactive uptake coefficient of $SO_2$ due to the oxidation by $NO_2$ based on sulfate production rate. We further assessed the importance of the multiphase oxidation of $SO_2$ by $NO_2$ in the atmosphere.

## 2 Experimental

### 2.1 Apparatus and procedures

The experimental setup used in this study is illustrated in Fig. 1. The details of the setup have been described previously (Liu et al., 2008; Zhao et al., 2011). $NO_2$ and $SO_2$ of certain concentrations were prepared by adjusting the flow rates of standard gases of specified concentrations ($NO_2$: 1000 ppm in $N_2$, Messer, Germany; $SO_2$: 2000 ppm in $N_2$, National Institute of Metrology P.R. China) and high-purity nitrogen (99.999%, Beijing

Haikeyuanchang Corp.). We used $N_2$ as a carrier gas to exclude the potential inference from other compounds in $SO_2$ oxidation such as $O_2$, which is key to investigate the direct oxidation of $SO_2$ by $NO_2$. RH was regulated by adjusting the flow rates of humidified $N_2$ and of dry $N_2$ and other dry gases. Humidified $N_2$ was prepared by bubbling $N_2$ through fritted glass in water. Flow rates of the gases were controlled by mass flow controllers (FC-260, Tylan, Germany). Mixed gases reacted with $CaCO_3$ particles in a stainless steel reaction cell. Individual $CaCO_3$ particles were deposited on a Teflon FEP film substrate annealed to a silicon wafer. The substrate was then placed in the reaction cell, which has a glass cover on top of the center. Through this top window, a Micro-Raman spectrometer (LabRam HR800, HORIBA Jobin Yvon) was used to acquire the Raman spectra of particles. A 514 nm excitation laser was focused onto selected particles and back scattering Raman signals were detected. The details of the instrument are described in previous studies (Liu et al., 2008; Zhao et al., 2011).

The RH and temperature of the outflow gas from the reaction cell were measured by a hygrometer (HMT100, Vaisala). Experiments of individual $CaCO_3$ particles reacting with $NO_2$ (75-200 ppm) and $SO_2$ (75-200 ppm) mixing gas diluted with $N_2$ were conducted under certain RH (17-72%). All the measurements were carried out at $25\pm0.5$ °C. Each reaction was repeated for three times.

In this study, the size of $CaCO_3$ particles was around 7-10 μm. During a reaction, components of an individual particle may distribute unevenly within the particle due to the formation of new aqueous phase or solid phase, and particles may grow. Because particles are larger than the laser spot (~1.5 μm), Raman spectrum from one point does not represent the chemical composition of the whole particle. Therefore Raman mapping was used to obtain the spectra on different points of a particle in order to get the chemical information of the whole particle. The mapping area is a rectangular slightly larger than the particle and mapping steps are 1×1 μm. Raman spectra in the range 800-3900 $cm^{-1}$ were acquired with exposure time of 1 s for each mapping point. During each mapping (7-10 min, depending on the mapping area), no noticeable change in composition was detected. The mean time of a mapping period was used as reaction time. During the reaction, microscopic images of particles were also recorded. Raman spectra were analyzed using Labspec 5 software (HORIBA Jobin Yvon). Raman peaks were fit to Gaussian-Lorentzian functions to obtain peak positions and peak areas on different points of the particle. The peak areas were then added up to get the peak area for the whole particle.

Besides the reaction of $CaCO_3$ with $SO_2$ and $NO_2$, other reaction systems including the reaction on $Ca(NO_3)_2$, $NaNO_3$, and $NH_4NO_3$ particles with $SO_2$ or $SO_2$ and $NO_2$ mixing gas (summarized in Table 1) were also studied in order to elucidate the reaction mechanism. Most experiments were conducted using $CaCO_3$ particles rather than directly using $Ca(NO_3)_2$ particles. $CaCO_3$ was selected because it is an important component of mineral aerosols especially in China as mentioned in the introduction and often used as a surrogate of mineral aerosols. Moreover, using $CaCO_3$ particles can better simulate the reaction on internally-mixed $CaCO_3$(solid)-$Ca(NO_3)_2$(aqueous) particles, which is widely observed in the ambient atmosphere and laboratory (Laskin et al., 2005; Zhang et al., 2003; Li and Shao, 2009; Sullivan et al., 2007; Li et al., 2010; Liu et al., 2008), and is formed via the reaction of $CaCO_3$ with acidic gases such as $HNO_3$ and $NO_2$ due to its alkalinity.

$CaCO_3$ (98%, Sigma) with diameters about 7-10 μm on average, $Ca(NO_3)_2 \cdot 4H_2O$ (ACS, 99-103%; Riedel-de Haën), $NH_4NO_3$ (AR, Beijing Chemical Works), and $NaNO_3$ (AR, Beijing Chemical Works) were used without further purification.

## 2.2 Quantification of reaction products on the particle phase

The Raman intensity of a sample is described as Equation (1):

$$I(\nu)=I_0 \cdot A(\nu) \cdot J(\nu) \cdot \nu^4 \cdot D \cdot K \tag{1}$$

where $I_0$ is the intensity of incident laser, $A(\nu)$ is the collection efficiency function of a Raman spectrometer, $J(\nu) \cdot \nu^4$ is the Raman scattering section of the sample, D is the number density of the sample, and K is the effective depth of the sample. Raman intensity is not only determined by the amount of the sample molecules, but also by the configuration of the instrument, whose influence cannot be eliminated unless internal standards are used. For soluble compounds, water can be used an internal standard (Zhao et al., 2011; Liu et al., 2008). However, in this study, one product ($CaSO_4$, see below) appeared as solid state. For solid particles of micro-scale, it is hard to add internal standards into the system. Therefore it is difficult to establish the relationship between Raman intensity and the amount of sample molecules, which makes the quantification very challenging.

In this study, we chose seven individual $CaSO_4$ particles varying in size as the standard for solid products. The profile of each particle can be obtained by scanning the particle using Raman mapping with steps of 1, 1, and 2 μm for x, y, and z dimension, respectively. The volume of each particle was calculated based on 3D profiles of the particles using a CAD software (AutoDesk). In order to minimize the influence of variations of incident laser on Raman intensity, these seven particles were measured before each experiment, which produced a calibration curve for each experiment (Fig. S1).

## 2.3 Determination of reactive uptake coefficient

In this study, sulfate was produced from the oxidation of $SO_2$. The reactive uptake coefficient $\gamma$ of $SO_2$ on individual particles was estimated from sulfate formation. $\gamma$ is derived as the rate of sulfate formation ($d\{SO_4^{2-}\}/dt$) divided by the rate of surface collisions with an individual particle (Z),

$$\gamma = \frac{\frac{d\{SO_4^{2-}\}}{dt}}{Z} . \tag{2}$$

$$Z = \frac{1}{4}cA_s[SO_2], \tag{3}$$

$$c = \sqrt{\frac{8RT}{\pi M_{SO_2}}} , \tag{4}$$

where R is the gas constant, T is temperature, $M_{SO_2}$ is the molecular weight of $SO_2$, and c is the mean molecular velocity of $SO_2$, $A_s$ is the surface area of an individual particle. Z is the collision rate between $SO_2$ and a particle. $\{SO_4^{2-}\}$ indicates the amount of sulfate on the particle phase in mole, and $[SO_2]$ indicates the concentration of $SO_2$ in the gas phase.

$\{SO_4^{2-}\}$ was determined by a calibration curve as stated above. In this study, since sulfate was mainly formed after the formation of $Ca(NO_3)_2$ droplet as shown below, $A_s$ was calculated by estimating the diameter of the droplet according to its microscopic image and using a shape of spherical segment defined by the contact angle of a water droplet on Teflon (Good and Koo, 1979). For each experiment, at least three particles with different diameters were measured to get an average reactive uptake coefficient.

# 3    Results and discussion

## 3.1  Reaction products and particle morphology changes

Figure 2 shows typical Raman spectra of a $CaCO_3$ particle during the reaction with $SO_2$ and $NO_2$. The peak at 1087 $cm^{-1}$ is assigned to the symmetric stretching mode of carbonate ($\nu_1$) (Nakamoto, 1997), which could be detected during the initial stage of the reaction. Shortly after the reaction started, a peak at 1050 $cm^{-1}$ was observed, which is attributed to the symmetric stretching mode of nitrate ($\nu_1$). This demonstrates that calcium nitrate ($Ca(NO_3)_2$) was produced during the reaction. A broad band at 2800-3800 $cm^{-1}$ was also observed together with the formation of $Ca(NO_3)_2$. It is assigned to –OH stretching of water in aqueous solution. The formation of aqueous solution is attributed to the deliquescence of $Ca(NO_3)_2$, which is very hygroscopic and can deliquesce at ~10% RH (Liu et al., 2008; Al-Abadleh et al., 2003; Tang and Fung, 1997). After about 82 min, a new peak at 1013 $cm^{-1}$ was observed, which is attributed to the symmetric stretching mode of sulfate ($\nu_1$) in anhydrite ($CaSO_4$) (Sarma et al., 1998). This peak clearly demonstrates that sulfate was formed. $CaSO_4$ as a reaction product has also been found in the reaction of $CaCO_3$ with $SO_2$ and $NO_2$ in a previous study (Ma et al., 2013b). Afterwards, no other Raman peaks than those of $CaCO_3$, $Ca(NO_3)_2$, and $CaSO_4$ were detected until 1050 min after the reaction.

Concomitant with the formation of $Ca(NO_3)_2$ and $CaSO_4$, the microscopic morphology of the particle changed significantly. The initial $CaCO_3$ particle was a crystal close to a rhombohedron of about 9-10 μm (Fig. 3a). After reacting with $NO_2/SO_2$, the surface of the particle became smoother, and then a liquid layer formed surrounding the solid particle core (Fig. 3c). Raman spectra of the particle reveal that the outer liquid layer consisted of $Ca(NO_3)_2$ and water. As the reaction proceeded, the solid $CaCO_3$ core diminished gradually and finally $CaCO_3$ completely disappeared and a $Ca(NO_3)_2$ spherical droplet was formed (Fig. 3d). The whole particle became larger due to the growth of the outer liquid layer. The diameter of the $Ca(NO_3)_2$ droplet reached ~16 μm, and the droplet did not change much in the subsequent period of the reaction. Despite the invariant droplet diameter, a new solid phase of needle-shaped crystals was formed as the reaction proceeded, which distributed unevenly in the droplet. The Raman spectra of the new solid phase and Raman mapping (Fig. S2) reveal that this solid matter was $CaSO_4$. The amount of $CaSO_4$ increased gradually during the reaction, and its Raman peak could be observed more clearly at 1050 min.

## 3.2  Reaction process

In order to learn about the reaction process and mechanism, the amounts of $Ca(NO_3)_2$, $CaSO_4$, and $CaCO_3$, represented by the peak area at 1050, 1013, and 1087 $cm^{-1}$ in Raman spectra, respectively, were investigated as a function of reaction time. As shown in Fig. 4, $Ca(NO_3)_2$ was produced before $CaSO_4$. Nitrate was detected immediately after the reaction started, and reached a maximum at ~50 min whereas sulfate did not reach the detection limit until 82 min of the reaction. Sulfate increased slowly in the reaction and we did not observe it leveling off even after 1050 min.

According to the time series of carbonate, nitrate, and sulfate, this reaction consisted of two successive processes. The first process was the formation of $Ca(NO_3)_2$, which was accompanied with the decline of $CaCO_3$ (Fig. 4), indicating that $Ca(NO_3)_2$ was produced due to the reaction of $CaCO_3$ with $NO_2$. $Ca(NO_3)_2$ has been

observed in the reaction of $CaCO_3$ with $NO_2$ in previous studies (Li et al., 2010; Tan et al., 2017). The formation of $Ca(NO_3)_2$ started with the reaction of $NO_2$ with adsorbed water or water in aqueous solution, forming $HNO_3$ and $HNO_2$. Then $HNO_3$ reacted with $CaCO_3$ forming $Ca(NO_3)_2$ as well as $CO_2$, which was released to the gas phase. $HNO_2$ could evaporate into the gas phase due to the continuous flushing of reactant gases during the experiments and acidity of the droplet (see below). The reaction equations are as follows:

$$NO_2(g) \leftrightarrow NO_2(aq) \tag{R1}$$

$$2NO_2(aq) + H_2O(aq) \longrightarrow HNO_3(aq) + HNO_2(aq) \tag{R2}$$

$$HNO_3(aq) \longrightarrow H^+(aq) + NO_3^-(aq) \tag{R3}$$

$$CaCO_3(s) + H^+(aq) \longrightarrow Ca^+(aq) + HCO_3^-(aq) \tag{R4}$$

$$HCO_3^-(aq) + H^+(aq) \longrightarrow H_2O(aq) + CO_2(g) \tag{R5}$$

$$HNO_2(aq) \leftrightarrow HNO_2(g) \tag{R6}$$

The detailed mechanism of the formation of $Ca(NO_3)_2$ in the reaction $CaCO_3$ with $NO_2$ have been studied by Li et al. (2010).

The second process was the formation of $CaSO_4$ through the oxidation of $SO_2$ by $NO_2$. $CaSO_4$ was mainly produced after $CaCO_3$ was completely reacted and increased steadily as the reaction proceeded. The amount of $Ca(NO_3)_2$ as the product of $NO_2$ uptake was overwhelmingly higher than that of $CaSO_4$ as the product of the reaction $SO_2$ with $NO_2$, which only reached detection limit after the complete conversion of $CaCO_3$. This indicates that the reaction of $SO_2$ with $NO_2$ did not contribute significantly to $NO_2$ uptake before $CaCO_3$ completely converted to $Ca(NO_3)_2$. Afterwards, the reaction of $SO_2$ with $NO_2$ promoted the reactive uptake of $NO_2$ by $Ca(NO_3)_2$ droplet.

## 3.3 Reaction mechanism

### 3.3.1 Mechanism of sulfate formation

Based on the results above, we found that a series of reactions of $SO_2$ and $NO_2$ on a $CaCO_3$ particle led to sulfate formation. Almost the entire sulfate was produced after a $CaCO_3$ particle was converted to a $Ca(NO_3)_2$ droplet (Fig. 4), although in some experiments a trace amount of sulfate could be observed when a small amount of $CaCO_3$ was still left in the $Ca(NO_3)_2$ droplet. The absence or low amount of sulfate before $CaCO_3$ was completely reacted might be due to the competition between the reaction of aqueous $NO_2$ with $CaCO_3$ and the reaction with $SO_2$. This result suggests that forming a $Ca(NO_3)_2$ droplet was key to the formation of sulfate.

This finding is further supported by the results of the reaction of $SO_2$ with $NO_2$ on a $Ca(NO_3)_2$ droplet (Fig. 5 and Table 1). Using a $Ca(NO_3)_2$ droplet as the reactant, the reaction with $SO_2/NO_2$ at the same condition still produced $CaSO_4$, confirming $CaCO_3$ was not necessary for sulfate formation. The reaction with $Ca(NO_3)_2$ produced similar amount of sulfate to the reaction with $CaCO_3$ based on Raman spectra and microscopic images (Fig. 5), which indicates that $Ca(NO_3)_2$ droplet was important for sulfate formation. Therefore, we conclude that $SO_2$ was mainly oxidized via the multiphase reaction on the $Ca(NO_3)_2$ droplet while $CaCO_3$ mainly worked as a precursor of the $Ca(NO_3)_2$ droplet.

The oxidant of $SO_2$ can be $NO_3^-$ or $NO_2$ in the $Ca(NO_3)_2$ droplet here. In a reaction between $Ca(NO_3)_2$ droplets and $SO_2$ (150 ppm) under 72% RH, we did not observe any sulfate formation on the basis of the Raman

spectra and microscopic image after 5 h of reaction. This indicates that $NO_3^-$ was not the oxidant for $SO_2$ in our study, which was also consistent with a previous study (Martin et al., 1981). Therefore, we conclude that $SO_2$ was oxidized by $NO_2$ in the $Ca(NO_3)_2$ droplet.

According to previous studies, $NO_2$ can oxidize sulfite and bisulfite ions into sulfate ion in aqueous phase (Ellison and Eckert, 1984; Shen and Rochelle, 1998; Littlejohn et al., 1993). The overall reaction equation was described to be (Clifton et al., 1988):

$$2NO_2(aq)+SO_3^{2-}(aq) + H_2O \longrightarrow 2H^+ + 2NO_2^-(aq) + SO_4^{2-}(aq) \qquad (R7)$$

$$2NO_2(aq)+HSO_3^-(aq) + H_2O \longrightarrow 3H^+ + 2NO_2^-(aq) + SO_4^{2-}(aq) \qquad (R8)$$

Under the experimental conditions of our study, water uptake of $Ca(NO_3)_2$ led to condensation of water, which provided a site for aqueous oxidation of S(IV) by $NO_2$. The relative fractions of the three S(IV) species depend on pH and the equilibrium between them is fast (Seinfeld and Pandis, 2006). The pH of the droplet was mainly determined by the gas-aqueous equilibrium of $SO_2$ in this study and estimated to be ~3. The characteristic time to reach the equilibrium in the gas-particle interface ($\sim10^{-5}$ s) was estimated to be much less than the characteristic time for the aqueous phase reaction of $SO_2$ with $NO_2$ ($10^{-2}$-$10^{-1}$ s) (Supplement S2). Therefore, aqueous S(IV) species can be considered to be in equilibrium with $SO_2$ in the gas phase. The concentrations of $HSO_3^-$, $H_2SO_3$, and $SO_3^{2-}$ were estimate to be ~$1.1\times10^{-3}$, $9.2\times10^{-5}$, and $6.6\times10^{-8}$ mol $L^{-1}$, respectively, using the equilibrium constants in Seinfeld and Pandis (2006) ($H_{SO2}$=1.23 mol $L^{-1}$ $atm^{-1}$, $K_1$=1.3×$10^{-2}$ mol $L^{-1}$, $K_2$=6.6×$10^{-8}$ mol $L^{-1}$) and thus the main S(IV) species was $HSO_3^-$. Then $SO_4^{2-}$ from S(IV) oxidation can react with $Ca^{2+}$ forming $CaSO_4$ precipitation as observed in Raman spectra due to the low value of $K_{sp}$ for $CaSO_4$ (Lide, 2009):

$$Ca^{2+}(aq)+SO_4^{2-}(aq)\rightleftharpoons CaSO_4(s) \qquad (R9)$$

Some previous studies have shown that $SO_2$ can react with $CaCO_3$ to produce calcium sulfite ($CaSO_3$) (Li et al., 2006; Prince et al., 2007b; Ma et al., 2013a), and $CaSO_3$ can be oxidized to $CaSO_4$ by $NO_2$ (Rosenberg and Grotta, 1980; Ma et al., 2013a). In our study, we investigated the reaction between $CaCO_3$ and $SO_2$ (150 ppm) at 72% RH. We found that both sulfate and sulfite were lower than the detection limit of our Raman spectrometer (~$5\times10^{-14}$ mol for sulfate at a signal to noise ratio of 2 and ~$3\times10^{-14}$ mol for sulfite according to the relative Raman scattering cross-section of sulfate and sulfite (Meyer et al., 1980)) even after 300 min of the reaction. This indicates that forming $CaSO_3$ was not the main pathway in $CaSO_4$ formation in our study and $CaCO_3$ did not directly contribute to the formation of $CaSO_4$.

### 3.3.2    Effects of cations in sulfate formation

Since sulfate was observed to precipitate as $CaSO_4$, we further analyzed the effect of precipitation reaction and cations on the aqueous oxidation of $SO_2$ by $NO_2$. In order to test effects of cations, we replaced $Ca^{2+}$ with $Na^+$ or $NH_4^+$. Based on Raman spectra, we found that in the reaction of a $NaNO_3$ or a $NH_4NO_3$ droplet with $NO_2/SO_2$, sulfate, either as aqueous ion (at 984 $cm^{-1}$ and 979 $cm^{-1}$ for $(NH_4)_2SO_4$ and $Na_2SO_4$, respectively) or as in $CaSO_4$ crystal, was below the detection limit after 300 min in the same reaction conditions as $Ca(NO_3)_2$ and $CaCO_3$ (Fig. 6 and Table 1). Considering that the Raman scattering cross-section of sulfate in $(NH_4)_2SO_4$ aqueous aerosol particle is even higher than sulfate in $CaSO_4$ (Wright, 1973; Stafford et al., 1976), it can be concluded that the sulfate production rate was larger in the presence of $Ca^{2+}$ compared to those in the presence of $Na^+$ or $NH_4^+$. The

difference can be explained by two possible reasons. The first possible reason may be due to the change of Gibbs energy. The spontaneity of the $SO_2$ oxidation by $NO_2$ for Reaction (R8) can be analyzed using the reaction Gibbs energy as follows:

$$\Delta_r G = \Delta_r G^\theta + RT ln \frac{a_{H^+}^3 \cdot a_{SO_4^{2-}} \cdot a_{NO_2^-}^2}{a_{NO_{2(aq)}}^2 \cdot a_{HSO_3^-}} \tag{5}$$

where $\Delta_r G$ is the reaction Gibbs energy, $\Delta_r G^\theta$ is the standard reaction Gibbs energy, R is the gas constant, T is temperature, and a is the activity of various species.

$\Delta_r G$ increases with increasing sulfate concentration. According to the different results between the reaction on $Ca(NO_3)_2$ droplet and the reaction on $NaNO_3$ and $NH_4NO_3$ droplet, there might be a backward reaction of $SO_2$ oxidation which consumed sulfate, although the detailed mechanism of the backward reaction is unknown at the moment. For $NaNO_3$ and $NH_4NO_3$ droplet, once sulfate concentration reached certain level, the reaction may stop due to the increase of $\Delta_r G$. For $Ca(NO_3)_2$ droplet, the precipitation of $CaSO_4$ can substantially decrease the activity of $SO_4^{2-}$, and thus decrease $\Delta_r G$ and promote the oxidation of $SO_2$ and sulfate formation. The second possible reason is that sulfate may crowd the reaction environment and suppress the colliding probability of S(IV) species with $NO_2$ in aqueous phase and the uptake coefficient of $SO_2$ or $NO_2$ on the droplet. Precipitation of sulfate as $CaSO_4$ can cancel such suppressions and thus promote the reaction. Regardless of the reasons behind, we can conclude that the precipitation of less soluble $CaSO_4$ promoted sulfate formation.

## 3.4 Reactive uptake coefficient of $SO_2$

The reactive uptake coefficients of $SO_2$ ($\gamma$) for sulfate formation under different conditions are shown in Table 2. Each reaction was repeated for three times, during which, three particles with different size were selected. $\gamma$ was higher at higher relative humidity, suggesting again that water in aqueous solution plays an important role in the formation of $CaSO_4$. At 17% RH, the reaction between $CaCO_3$ and $NO_2$ (the first process of the whole reaction) proceeded very slowly, and the amount of water in aqueous solution formed due to the water uptake of $Ca(NO_3)_2$ was very low. As a result, we did not observe the formation of $CaSO_4$ (the second process of the whole reaction) after 1000 min of the reaction and even at higher $SO_2$ and $NO_2$ concentrations (200 ppm $SO_2$, 200 ppm $NO_2$). Under higher relative humidity (46% and 72% RH), sulfate was observed soon after the reaction. It is interesting to note that there were no significant difference for $\gamma$ between 46% and 72% RH. In either case, the reaction between $CaCO_3$ and $NO_2$ proceeded quickly and $CaCO_3$ was completely converted to a $Ca(NO_3)_2$ droplet within 100 min after the reaction. In the presence of enough water in aqueous solution, RH seemed to be no longer a limiting factor. In such conditions, an increase of $NO_2$ concentration (from 75 ppm to 200 ppm at 72% RH) promoted the reactive uptake of $SO_2$.

The reactive uptake coefficient of $SO_2$ for sulfate formation was determined to be on the order of $10^{-8}$ at 46% and 72% RH. This value is higher than the uptake coefficient ($10^{-10}$) on mineral particles sampled from Cape Verde Islands (the main contents being potassium feldspars and quartz) obtained by Ullerstam et al. (2003) using $NO_2/SO_2$ mixing gas and Diffuse Reflectance Infrared Fourier Transform Spectroscopy (DRIFTS) technique. But the uptake coefficient in this study is lower than the uptake coefficient of $SO_2$ on Arizona Test Dust (ATD) particles in the presence of $NO_2$ (($2.10\pm0.08$)$\times10^{-6}$) determined by Park and Jang (2016). $\gamma$ here is also much lower than the $\gamma$ of $SO_2$ on oxalic acid particles in the presence of $NO_2$ and $NH_3$ ($10^{-6}\sim10^{-4}$) determined at varying

RH reported by Wang et al. (2016). The difference in these uptake coefficients is attributed to the different chemical composition of particles, reaction mechanism, reaction conditions, and the ways that the particle surface is determined. It is worth noting that in the studies of Ullerstam et al. (2003) and Park and Jang (2016), particles exist as solid state and sulfate formation is via gas-solid heterogeneous reaction, and in the study of Wang et al. (2016) sulfate formation is stated to be via aqueous reaction. In this study sulfate formation was via gas-liquid-solid multiphase reaction and water in aqueous solution played a key role.

The $\gamma$ of $SO_2$ was further compared with the reaction rate constants of the aqueous reaction of $NO_2$ with sulfite and bisulfite in bulk solution in the literature by deriving $\gamma$ from rate constants using the method in Davidovits et al. (2006). The detailed method can be referred to the supplement S1. Lee and Schwartz (1983) determined the rate constant of the reaction of $NO_2$ with bisulfite to be $>2\times10^6$ mol$^{-1}$ L s$^{-1}$ at pH 5.8 and 6.4. Clifton et al. (1988) determined the rate constant of the reaction of $NO_2$ with sulfite/bisulfite to be $(1.24\text{-}2.95)\times10^7$ mol$^{-1}$ L s$^{-1}$ at pH 5.6-13 and further reported a rate constant of $1.4\times10^5$ mol$^{-1}$ L s$^{-1}$ at pH 5 from the study of Lee and Schwartz (1983). The different rate constants were attributed to the different approaches to determine the reaction rate by Clifton et al. (1988). Clifton et al. (1988) determined the reaction rate from the consumption rate the reactant, $NO_2$, which corresponds to the first reaction step of $NO_2$ with S(IV). Yet, Lee and Schwartz (1983) determined the reaction rate from the production rate of products (their conductivity), which is expected to be much slower than $NO_2$ consumption since formation of products needs more steps. In this study, we determined $\gamma$ using sulfate production rate, and thus our data are comparable to the study of Lee and Schwartz (1983). Yet, the study of Lee and Schwartz (1983) only covers a pH range of 5-6.4 and has no overlap with the pH (~3) in our study, therefore uptake coefficients from both studies are not directly comparable. Nevertheless, the reaction rate of $1.4\times10^5$ mol$^{-1}$ L s$^{-1}$ at pH 5 corresponds to the uptake coefficient of $4.3\times10^{-7}$, which is around one order of magnitude higher than the uptake coefficient in our study determined at pH ~3 for the droplet. The difference may be due to the different pH between these two studies, the different mechanisms between the multiphase reaction on particles and bulk aqueous reaction, and the different concentrations of each S(IV) species since the different species may have different reactivity with $NO_2$. The reaction rate of S(IV) has been found to decrease with decreasing pH and the reactivity of sulfite with $NO_2$ seems to be higher than bisulfite (Lee and Schwartz, 1983; Clifton et al., 1988; Takeuchi et al., 1977). In addition, the ionic strength in the droplet of this study (15-55 mol Kg$^{-1}$) was much higher than that in the bulk solution in previous studies (on the order of $10^{-6}$-$10^{-1}$ mol Kg$^{-1}$), which may also influence the reaction rate.

In the ambient atmosphere, the reactive uptake coefficient of $SO_2$ due to the multiphase oxidation by $NO_2$ is influenced by various factors such as RH, $NO_2$ concentration, pH, sulfate concentration, and the presence of other ions in aerosol particles. For example, $NO_2$ concentrations in the atmosphere are much lower than those used in this study. At lower $NO_2$ concentrations, the uptake coefficient of $SO_2$ decreases, because the oxidation rate of $SO_2$ in aqueous phase decreases with decreasing $NO_2$ concentration. In addition, aqueous sulfate concentrations in aerosol particles in the atmosphere are often high. According to the effect of cations (Section 3.3.2), while reduced sulfate concentration by $CaSO_4$ precipitation likely led to the enhanced sulfate production rate in the reaction of $SO_2$ on $Ca(NO_3)_2$, higher sulfate concentration could increase the reaction Gibbs energy $\Delta_rG$ (as shown in Eq. 5) and reduce the colliding probability of S(IV) species with $NO_2$ in the aqueous phase as discussed above and thus suppress the reaction of $SO_2$ and $NO_2$. This can reduce the uptake coefficient of $SO_2$. Therefore,

the reactive uptake coefficient of $SO_2$ obtained in this study ($10^{-8}$ at 46-72% RH and 75 ppm $NO_2$) can be
regarded as an upper limit of the reactive uptake coefficient of $SO_2$ due to the multiphase reaction with $NO_2$ in
the ambient atmosphere.

## 357 4     Conclusion and implications

We investigated the heterogeneous reaction of $SO_2$ directly with $NO_2$ on individual $CaCO_3$ particles in $N_2$
using Micro-Raman spectrometry. The reaction first converted the $CaCO_3$ particle to the $Ca(NO_3)_2$ droplet via the
reaction with $NO_2$ in the $SO_2/NO_2/H_2O/N_2$ gas mixture and the deliquescence of $Ca(NO_3)_2$, and then formed
needle-shaped $CaSO_4$ crystals in the $Ca(NO_3)_2$ droplet via the multiphase reaction of $SO_2$ with $NO_2$. The sulfate
formation was observed only during the multiphase oxidation by $NO_2$, that is, after the complete conversion of
$CaCO_3$ to $Ca(NO_3)_2$ droplet. The precipitation of $CaSO_4$ from solution promoted sulfate formation. The reactive
uptake coefficient of $SO_2$ for sulfate formation in the multiphase reaction with $NO_2$ is on the order of $10^{-8}$ under
the experimental conditions of this study (RH: 46-72%, $NO_2$: 75 ppm). The reactive uptake coefficient of $SO_2$
was found to be enhanced at higher RH.
In order to assess the importance of the multiphase reaction of $SO_2$ directly oxidized by $NO_2$ to sulfate in the
atmosphere, we compare the lifetime of $SO_2$ due to the multiphase oxidation of $SO_2$ by $NO_2$ with the lifetime due
to the gas phase oxidation of $SO_2$ by OH. Using a daytime OH concentration of $1 \times 10^6$ molecule $cm^{-3}$ (Lelieveld
et al., 2016; Prinn et al., 2005), the lifetime of $SO_2$ in the atmosphere due to gas phase OH oxidation is around 12
days. The life time of $SO_2$ due to the multiphase oxidation by $NO_2$ is around 7000 days using the uptake
coefficient of $SO_2$ from this study ($3.22 \times 10^{-8}$) and a typical particle surface area concentration for mineral
aerosols in winter in Beijing ($6.3 \times 10^{-6}$ $cm^2$ $cm^{-3}$) (Huang et al., 2015). Using an annual average particle surface
area concentration of $PM_{10}$ in Beijing ($1.4 \times 10^{-5}$ $cm^2$ $cm^{-3}$)(Wehner et al., 2008) results in a $SO_2$ life time of 3300
days due to the multiphase oxidation by $NO_2$. In the atmosphere, the lifetime of $SO_2$ due to the multiphase
oxidation by $NO_2$ should be even longer than these values because the uptake coefficient of $SO_2$ used here
($3.22 \times 10^{-8}$) is an upper limit of the uptake coefficient of $SO_2$ in the ambient atmosphere as discussed above. This
comparison indicates that the direct multiphase oxidation of $SO_2$ by $NO_2$ is unlikely to be an important sink of
$SO_2$ and source of sulfate compared with the oxidation of $SO_2$ by OH.
It is worth mentioning that this study did not investigate the dependence of the reactive uptake coefficient
due to the direct oxidation of $SO_2$ by $NO_2$ on pH, especially not under high pH conditions, for which recent
studies have claimed this reaction to be important (Cheng et al., 2016; Wang et al., 2016). Because of the
important role of multiphase/heterogeneous reactions in $SO_2$ oxidation found in the atmosphere and the low
reaction rate of the direct multiphase oxidation of $SO_2$ by $NO_2$, it is more likely that the aqueous reactions of $SO_2$
with other oxidants, such as the reactions with $H_2O_2$, $O_3$, and $O_2$, with or without transition metals, could be
important pathways for sulfate formation in the atmosphere.
**Acknowledgements**
This work was supported by Natural Science Foundation Committee of China (41421064, 21190051, and
40490265) and Ministry of Science and Technology (Grant No. 2002CB410802).

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

Table 1 Summary of the results obtained in different reaction systems

| Particle | Gases | RH (%) | Whether sulfate was detected |
|---|---|---|---|
| $CaCO_3$ | $SO_2$(75 ppm)+$NO_2$(75 ppm) | 72 | Yes |
| $Ca(NO_3)_2$ droplet | $SO_2$(75 ppm)+$NO_2$(75 ppm) | 72 | Yes |
| $CaCO_3$ | $SO_2$ (150 ppm) | 72 | No |
| $Ca(NO_3)_2$ droplet | $SO_2$ (150 ppm)) | 72 | No |
| $NaNO_3$ droplet | $SO_2$(75 ppm)+$NO_2$(75 ppm) | 72 | No |
| $NH_4NO_3$ droplet | $SO_2$(75 ppm)+$NO_2$(75 ppm) | 72 | No |


Table 2. Reactive uptake coefficient of $SO_2$ for sulfate formation ($\gamma$) during the reaction of $SO_2$ with $NO_2$ on
individual $CaCO_3$ particles under different conditions at 298 K.

| [$SO_2$] (ppm) | [$NO_2$] (ppm) | RH (%) | $\gamma$ ($\times 10^{-8}$) |
|---|---|---|---|
| 75 | 75 | 72 | 3.22±1.08 [b] |
| 75 | 200 | 72 | 16.0±3.12 |
| 75 | 75 | 46 | 3.22±0.90 |
| 75 | 75 | 17 | 0 [a] |
| 200 | 200 | 17 | 0 [a] |

[a]: Sulfate was below the detection limit.
[b]: The uncertainties are the standard deviations of $\gamma$ from duplicate experiments.

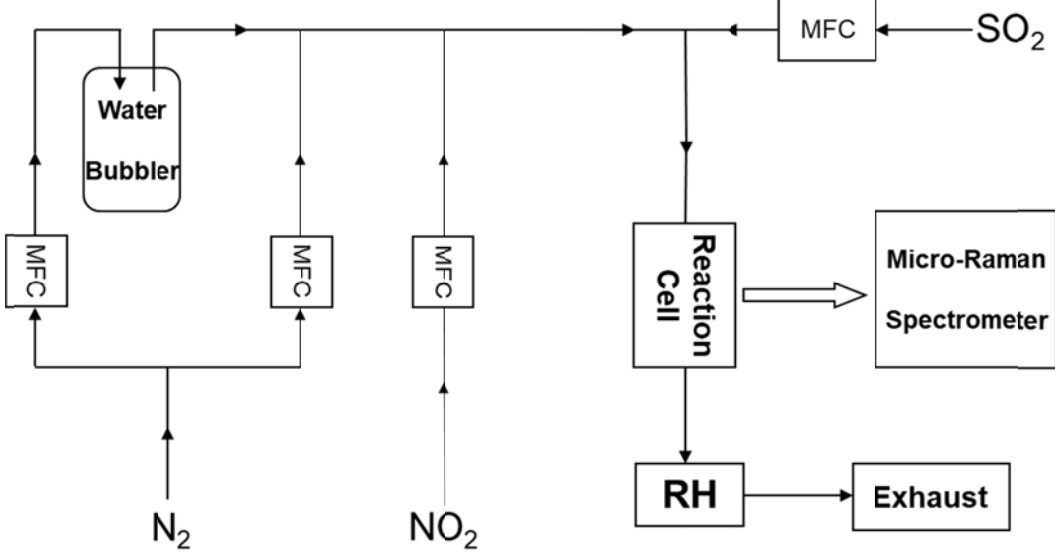


Fig. 1. Schematic diagram of the experimental setup. MFC: mass flow controller.

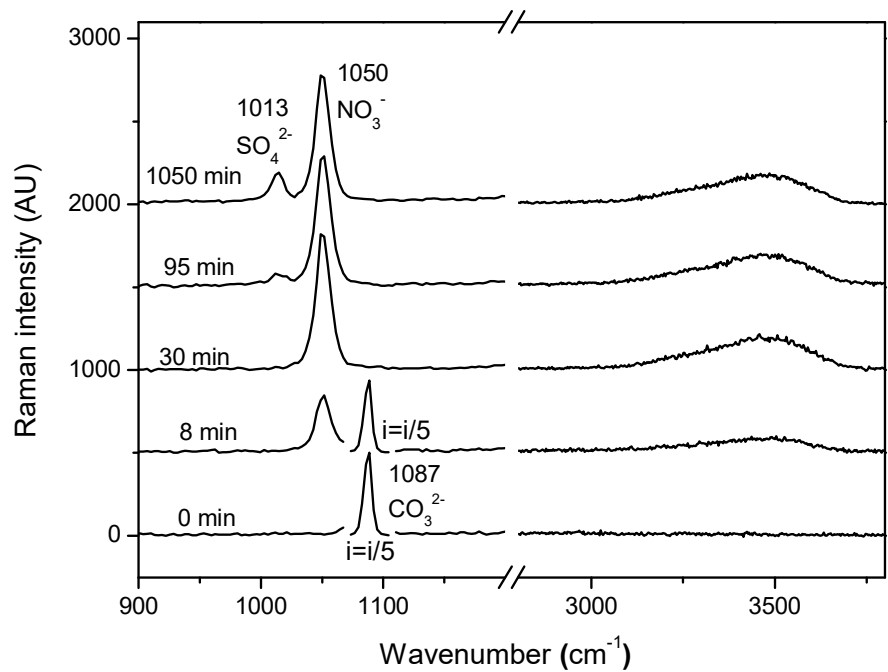


Fig. 2. Raman spectra of an individual $CaCO_3$ particle during the reaction with $NO_2$ (75 ppm) and $SO_2$ (75 ppm)
at 72% RH at the reaction time of 0, 8, 30, 95, and 1050 min.

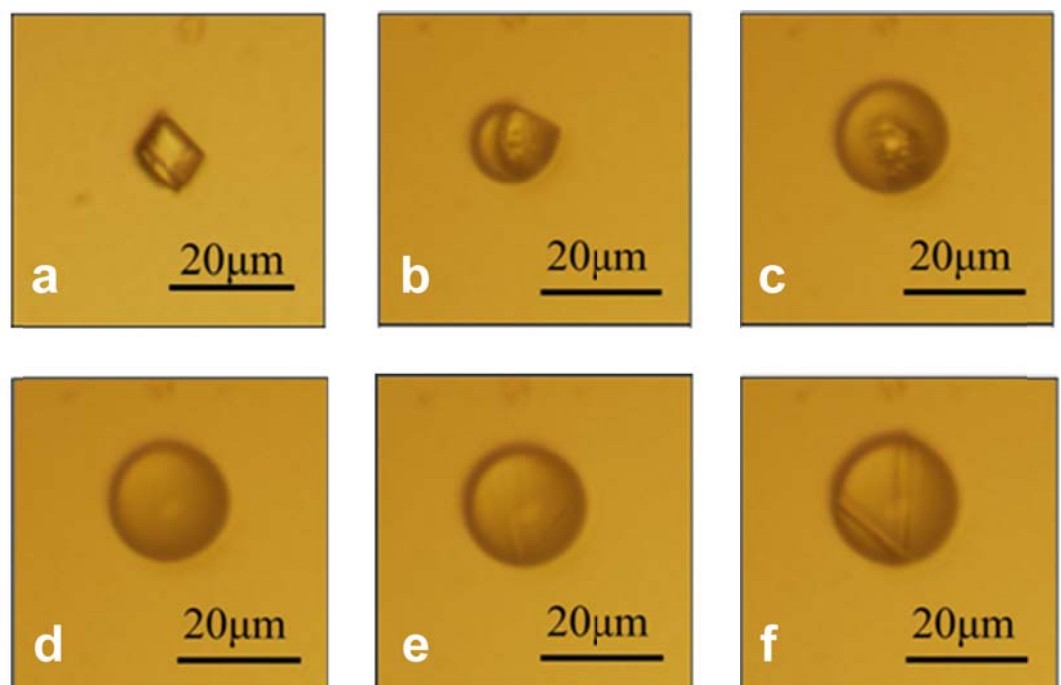

Fig. 3. Microscopic images of an individual CaCO$_3$ particle (same as in Fig. 2) reacting with NO$_2$ (75 ppm) and
SO$_2$ (75 ppm) at 72% RH. a-f corresponds to the reaction time of 0, 6, 29, 37, 94, and 1050 min, respectively.

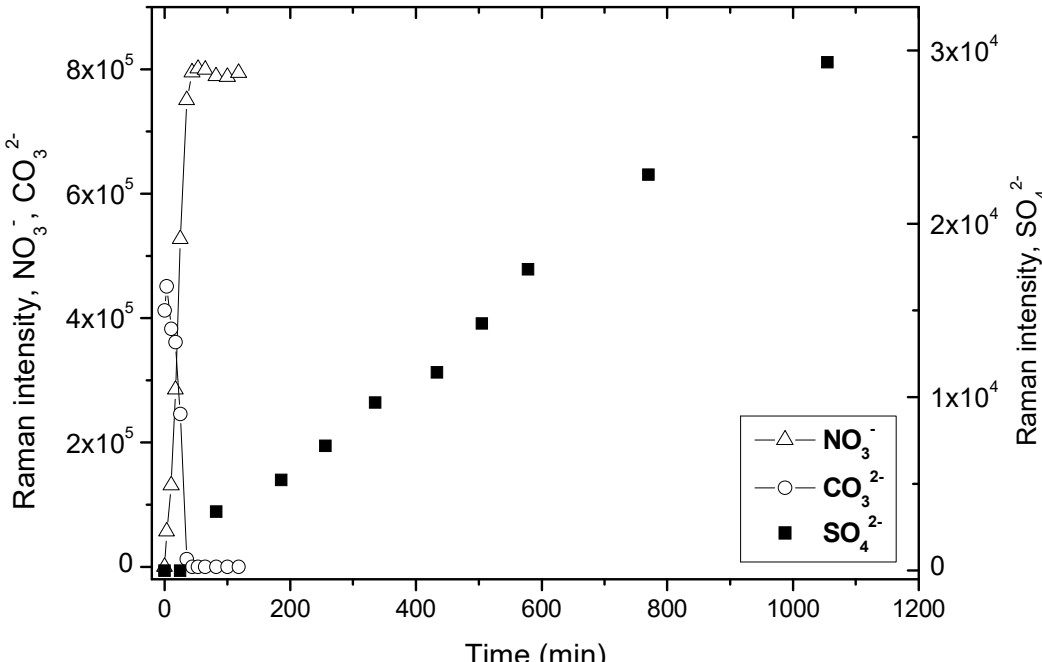


Fig. 4. Raman peak intensity of carbonate, nitrate (left axis), and sulfate (right axis) as a function of time during
the reaction of an individual $CaCO_3$ particle with $NO_2$ (75ppm) and $SO_2$ (75ppm) at 72% RH (same as in Fig. 2
and 3). Note that the scales of the left axis and right axis are different. The intensity of $NO_3^-$, $SO_4^{2-}$, and $CO_3^{2-}$
show the peak area at 1050, 1013, and 1087 $cm^{-1}$, respectively, in Raman spectra obtained by Raman mapping.
By 118 min, $CaCO_3$ was completely converted to $Ca(NO_3)_2$. Carbonate had decreased to zero and nitrate had
reached a plateau. Therefore no further data of carbonate and nitrate were shown.


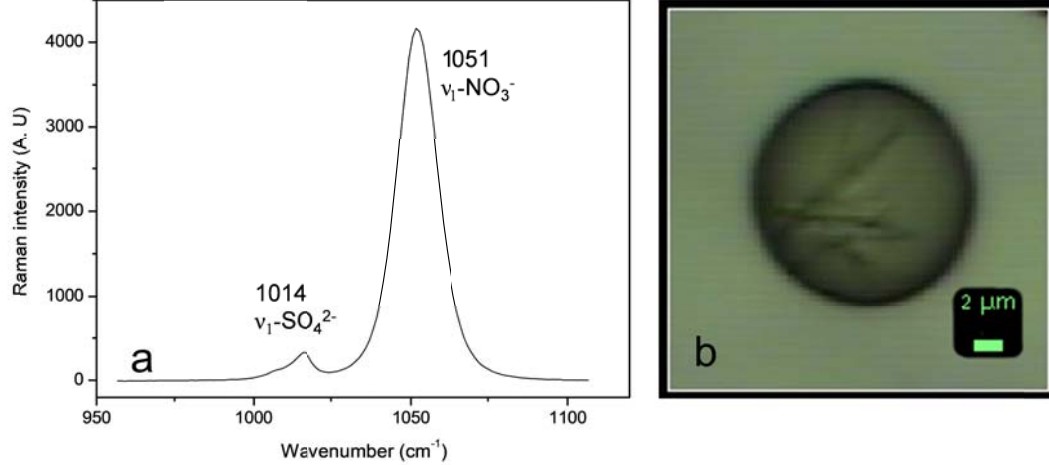


Fig. 5. Raman spectra (a) and microscopic image (b) of a Ca(NO$_3$)$_2$ droplet reacting with NO$_2$ (75 ppm) and SO$_2$
(75 ppm) at 72% RH at a reaction time of 300 min. The peak at 1014 cm$^{-1}$ in Raman spectra and crystals from the
microscopic image indicate CaSO$_4$ was formed in this reaction.

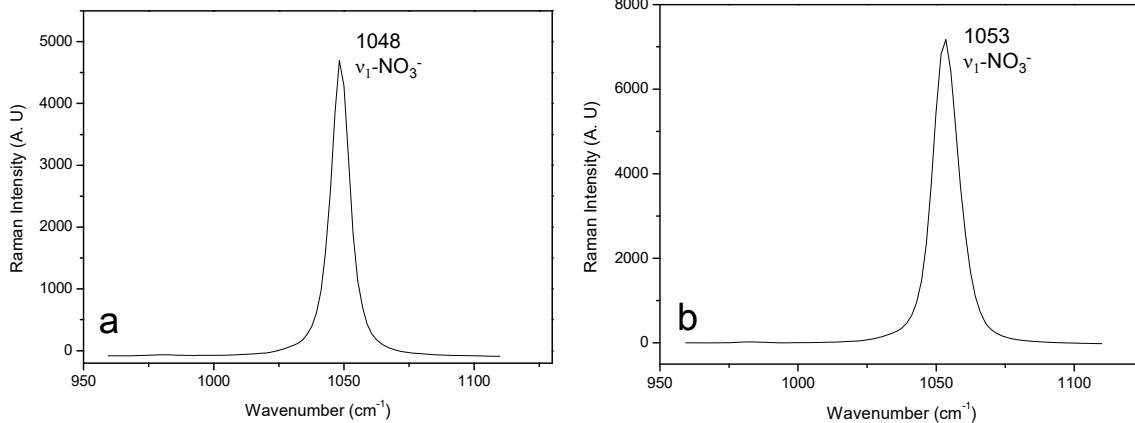


Fig. 6. Raman spectra of a $NH_4NO_3$ (a) and $NaNO_3$ (b) droplet reacting with $NO_2$ (75 ppm) and $SO_2$ (75 ppm) at
72% RH at the reaction time of 300 min.