# Peer review of "Multiphase Oxidation of SO2 by NO2 on CaCO3 Particles"

_Atmospheric Chemistry and Physics, 2017_

## Referee Comment (RC1) · Anonymous Referee #1 · 19 Jul 2017

General comments

The paper presents the results on multiphase reaction of SO2 with NO2 on individual CaCO3 particles in N2 at RH between 17 and 72% using Micro-Raman spectrometer with a flow reaction system. The reaction process was systematically investigated and found that CaCO3 converts first to Ca(NO3)2 (by the reaction with NO2) and its deliquescence to droplet, where further SO2 oxidizes with NO2 forming CaSO4. The reactive uptake coefficient $\gamma$ of SO2 was determined on the basis of sulfate formation rate.

Although many studies concerning SO2 oxidation in the atmosphere were performed in late 1980s and 1990s, mainly due to much higher pollution with SO2 all over the world at that time, and various pathways of oxidation were identified, the questions

[Figure]

[Figure]

The paper presents the results on multiphase reaction of SO2 with NO2 on individual CaCO3 particles in N2 at RH between 17 and 72% using Micro-Raman spectrometer with a flow reaction system. The reaction process was systematically investigated and found that CaCO3 converts first to Ca(NO3)2 (by the reaction with NO2) and its deliquescence to droplet, where further SO2 oxidizes with NO2 forming CaSO4. The reactive uptake coefficient $\gamma$ of SO2 was determined on the basis of sulfate formation rate.

Although many studies concerning SO2 oxidation in the atmosphere were performed in late 1980s and 1990s, mainly due to much higher pollution with SO2 all over the world at that time, and various pathways of oxidation were identified, the questions

concerning sulfate formation have not yet been fully resolved. As shown e.g., that high concentrations of sulfate during heavily polluted episodes under haze conditions in China could not be explained on the basis of known pathways only. In addition, due to the progress and development of techniques, nowadays there are more possibilities to study processes also on the micro level as shown in this paper.

Thus, I found the paper of sufficient atmospheric interest to merit publication after revision; in "specific comments" some questions and/or comments are listed which should be considered.

However, I would strongly recommend showing also the results for the system SO2/NO2/H2O/O2 together with those presented here and not in the next paper as said in line 80. Experimental conditions will be closer to atmospheric, and as mentioned on p.9 (lines 301-303) it is expected that the reactions under O2 are faster and could be more important source of sulfate.

Specific comments Introduction: (1) P.3, lines 59-62: It would be worth to mention also the studies from 2001 (Turšič et al. 2001, Atmos. Environ.).

Experimental: (2) The experimental approach (Raman mapping analysis) where you can follow the changes during the reaction course (as can be seen in Fig. S2) is intriguing.

Results and discussion: (3) Fig. 2 nicely shows how the peak for CO32- decreases and disappears after certain time of reaction; the change can be seen also in Fig. 3. What happens to it (releases as CO2)? (4) It is not correct to explain its "disappearance" as "completely consumed" (line 171).

(5) Line 186: "the consumption of CaCO3" is not appropriate

(6) Lines 186-187: If Ca(NO3)2 is formed in the reaction between CaCO3 and NO2, NO2 should first disproportionate to NO3- and NO2-, which is possible in the presence of water. How is then Ca(NO3)2 first formed from CaCO3, and only then converts into

droplet in the presence of water? The authors should explain the reactions also for the first step, i.e. the conversion of CaCO3 to Ca(NO3)2 although the reference is given (line 188). I suggest that the complete mechanism is written.

(7) The authors may want to add a reference of Tan et al., 2016, ACP.

(8) It is concluded that aqueous phase plays a key role in SO2 oxidation by NO2, which is also known from previous studies. Line 219: pH is estimated to be around 3. What would be the concentrations of reactive species in Ca(NO3)2 droplet?

(9) Lines 236-241: This part is not well understandable. It is concluded that precipitation of CaSO4 formed in/on Ca(NO3)2 droplet promotes sulfate formation. On the other hand, when NaNO3 or NH4NO3 droplet is used instead of Ca(NO3)2, no sulfate was formed after 300 min. If aqueous phase is a key factor for the oxidation of SO2 with NO2, then this should happen also in these droplets, although the reaction is most probably much slower. Why the reaction was not carried out at longer times?

(10) Line 240: In droplets of NaNO3 or NH4NO3, CaSO4 cannot be formed.

(11) Line 250: Is it correct that at RH of 46% the conditions for a complete conversion into a Ca(NO3)2 droplet are achieved?

(12) Line 259: Write what is DRIFTS technique (it was not mentioned before).

(13) Line 206: ATD particles?

(14) Lines 273-275: Is this statement correct? Higher concentrations of aqueous sulfate may suppress the reaction between SO2 and NO2, while CaSO4 precipitation can promote it.

Please also note the supplement to this comment:
https://www.atmos-chem-phys-discuss.net/acp-2017-610/acp-2017-610-RC1-supplement.pdf

**Supplement:**

**General comments**

The paper presents the results on multiphase reaction of $SO_2$ with $NO_2$ on individual $CaCO_3$ particles in $N_2$ at RH between 17 and 72% using Micro-Raman spectrometer with a flow reaction system. The reaction process was systematically investigated and found that $CaCO_3$ converts first to $Ca(NO_3)_2$ (by the reaction with $NO_2$) and its deliquescence to droplet, where further $SO_2$ oxidizes with $NO_2$ forming $CaSO_4$. The reactive uptake coefficient γ of $SO_2$ was determined on the basis of sulfate formation rate.

Although many studies concerning $SO_2$ oxidation in the atmosphere were performed in late 1980s and 1990s, mainly due to much higher pollution with $SO_2$ all over the world at that time, and various pathways of oxidation were identified, the questions concerning sulfate formation have not yet been fully resolved. As shown e.g., that high concentrations of sulfate during heavily polluted episodes under haze conditions in China could not be explained on the basis of known pathways only. In addition, due to the progress and development of techniques, nowadays there are more possibilities to study processes also on the micro level as shown in this paper.

Thus, I found the paper of sufficient atmospheric interest to merit publication after revision; in "specific comments" some questions and/or comments are listed which should be considered.

However, I would strongly recommend showing also the results for the system $SO_2/NO_2/H_2O/O_2$ together with those presented here and not in the next paper as said in line 80. Experimental conditions will be closer to atmospheric, and as mentioned on p.9 (lines 301-303) it is expected that the reactions under $O_2$ are faster and could be more important source of sulfate.

**Specific comments**

Introduction:
(1) P.3, lines 59-62: It would be worth to mention also the studies from 2001 (Turšič et al. 2001, Atmos. Environ.).

Experimental:
(2) The experimental approach (Raman mapping analysis) where you can follow the changes during the reaction course (as can be seen in Fig. S2) is intriguing.

Results and discussion:
(3) Fig. 2 nicely shows how the peak for $CO_3^{2-}$ decreases and disappears after certain time of reaction; the change can be seen also in Fig. 3. What happens to it (releases as $CO_2$)?
(4) It is not correct to explain its "disappearance" as "completely consumed" (line 171).

(5) Line 186: "the consumption of $CaCO_3$" is not appropriate

(6) Lines 186-187: If $Ca(NO_3)_2$ is formed in the reaction between $CaCO_3$ and $NO_2$, $NO_2$ should first disproportionate to $NO_3^-$ and $NO_2^-$, which is possible in the presence of water. How is then $Ca(NO_3)_2$ first formed from $CaCO_3$, and only then converts into droplet in the presence of water? The authors should explain the reactions also for the first step, i.e. the conversion of $CaCO_3$ to $Ca(NO_3)_2$ although the reference is given (line 188). I suggest that the complete mechanism is written.

(7) The authors may want to add a reference of Tan et al., 2016, ACP.

(8) It is concluded that aqueous phase plays a key role in $SO_2$ oxidation by $NO_2$, which is also known from previous studies. Line 219: pH is estimated to be around 3. What would be the concentrations of reactive species in $Ca(NO_3)_2$ droplet?

(9) Lines 236-241: This part is not well understandable. It is concluded that precipitation of $CaSO_4$ formed in/on $Ca(NO_3)_2$ droplet promotes sulfate formation. On the other hand, when $NaNO_3$ or $NH_4NO_3$ droplet is used instead of $Ca(NO_3)_2$, no sulfate was formed after 300 min. If aqueous phase is a key factor for the oxidation of $SO_2$ with $NO_2$, then this should happen also in these droplets, although the reaction is most probably much slower. Why the reaction was not carried out at longer times?

(10) Line 240: In droplets of $NaNO_3$ or $NH_4NO_3$, $CaSO_4$ cannot be formed.

(11) Line 250: Is it correct that at RH of 46% the conditions for a complete conversion into a $Ca(NO_3)_2$ droplet are achieved?

(12) Line 259: Write what is DRIFTS technique (it was not mentioned before).

(13) Line 206: ATD particles?

(14) Lines 273-275: Is this statement correct? Higher concentrations of aqueous sulfate may suppress the reaction between $SO_2$ and $NO_2$, while $CaSO_4$ precipitation can promote it.

---

## Referee Comment (RC2) · Anonymous Referee #2 · 17 Aug 2017

This study investigated the heterogeneous reaction of SO2 with NO2 on individual CaCO3 particles in N2 using Micro-Raman spectroscopy. The results show that CaCO3 was first converted to Ca(NO3)2 forming a droplet and promoting the oxidation of SO2 by NO2.The precipitation of CaSO4 was suggested as a key step accelerating the sulfate formation. Based on the uptake coefficient determined, the authors concluded that the SO2 + NO2 reaction was not important compared to the oxidation of SO2 by OH radicals. The experiment was well designed and the paper was well written. But I do have concerns about the role of CaSO4 precipitation and I would also suggest the authors to compare their results with literature data before making strong statement on the role of NO2+SO2 chemistry.

Major concern:

[Figure]

1. The authors generalized the results of their CaCO3 experiments to assess the role of NO2+SO2 chemistry. I am not sure if such generalization is correct because according to early studies of Lee and Schwartz,1983 and Clifton et al.,1988, this reaction can be important under polluted and less acidic conditions in contrary to the authors' statement. The authors used deposited super-micro particles in their experiments. But I don't expect much difference between such a system and bulk experiments because large particles are not subject to strong Kelvin effect and particles contacted with substrates would not become supersaturated solution of high ionic strength due to nucleation. Thus before generalizing results for ambient aerosols, I would suggest the authors to discuss their difference with those early studies.

2. Based on Equation (5), the authors concluded that the precipitation-induced reduction of sulfate will promote the oxidation of SO2 by NO2 (reaction 2). I don't know if it is correct to use Eq. (5) in this way. Because Equation (5) is valid for reversible reactions and removing/adding products of non-reversible reactions will not change the reaction rate much.

Other comments:

Page 5 line 133, half sentene?

Page 6 line 187, I would suggest to briefly describe the mechanism of Ca(NO3)2 formation. Will the present of SO2 influence the uptake of NO2?

Fig. 4, no data for nitrate and carbonate after 120 min, why?

---

## Author Comment (AC1) · 10 Oct 2017

**Responses to Referee # 1**

We thank the reviewer for carefully reviewing our manuscript. The comments and suggestions are greatly appreciated. All the comments have been addressed. In the following, please find our responses to the comments one by one and corresponding revisions made to the manuscript. The original comments are shown in italics. The revised parts of the manuscript are highlighted.

*General comments*

*The paper presents the results on multiphase reaction of $SO_2$ with $NO_2$ on individual $CaCO_3$ particles in $N_2$ at RH between 17 and 72% using Micro-Raman spectrometer with a flow reaction system. The reaction process was systematically investigated and found that $CaCO_3$ converts first to $Ca(NO_3)_2$ (by the reaction with $NO_2$) and its deliquescence to droplet, where further $SO_2$ oxidizes with $NO_2$ forming $CaSO_4$. The reactive uptake coefficient $\gamma$ of $SO_2$ was determined on the basis of sulfate formation rate.*

*Although many studies concerning $SO_2$ oxidation in the atmosphere were performed in late 1980s and 1990s, mainly due to much higher pollution with $SO_2$ all over the world at that time, and various pathways of oxidation were identified, the questions concerning sulfate formation have not yet been fully resolved. As shown e.g., that high concentrations of sulfate during heavily polluted episodes under haze conditions in China could not be explained on the basis of known pathways only. In addition, due to the progress and development of techniques, nowadays there are more possibilities to study processes also on the micro level as shown in this paper.*

*Thus, I found the paper of sufficient atmospheric interest to merit publication after revision; in "specific comments" some questions and/or comments are listed which should be considered.*

*However, I would strongly recommend showing also the results for the system $SO_2/NO_2/H_2O/O_2$ together with those presented here and not in the next paper as said in line 80. Experimental conditions will be closer to atmospheric, and as mentioned on p.9 (lines 301-303) it is expected that the reactions under O2 are faster and could be more important source of sulfate.*

**Response:**

We thank the reviewer for the supporting remarks.

As to the recommendation "*showing also the results for the system $SO_2/NO_2/H_2O/O_2$ together with those presented here and not in the next paper as said in line 80*", we realized that our phrasing in the some texts of the manuscript was not precise and clear enough to express our primary motivation. We have modified these texts (e.g. lines 80-83, lines 350-357) in the revised manuscript to clearly state our motivation, i.e., to address the multiphase reaction of $SO_2$ directly with $NO_2$ and evaluate the

importance of this reaction pathway in sulfate formation in the real atmosphere, which was proposed by a number of recent studies (Cheng et al., 2016; Wang et al., 2016; Xue et al., 2016) but remains unclear. In order to exclude potential confounding reactions from other compounds, we used inert $N_2$ as a carrier gas. For this motivation, the direct reaction of $SO_2$ with $NO_2$ has ambient relevance no matter whether $O_2$ is present or not.

The reason that we did not include the multiphase reaction of $SO_2$ with $O_2/NO_2$ in this paper is because we found that in this reaction $SO_2$ was actually oxidized by $O_2$, not by $NO_2$. It is distinct from the reaction of $SO_2$ directly with $NO_2$ with markedly different mechanisms, products, and atmospheric implications, as we will show (Yu et al., 2017). $O_2$ was the main oxidant of $SO_2$ and $NO_2$ only acted as an initiator of chain reactions. The atmospheric implications are significantly different from the direct reaction of $SO_2$ with $NO_2$ because not only the oxidation of $SO_2$ by $O_2$ leads to much faster sulfate oxidation but also the reaction is not linked to reactive nitrogen chemistry in the atmosphere. Therefore, we address this reaction in a separate companion manuscript (Yu et al., 2017).

In addition, in order to reflect the distinction of these two studies more precisely, we have revised the title of our manuscript as follows:

"Multiphase Reaction of $SO_2$ on $CaCO_3$ Particles. 1. Oxidation of $SO_2$ by $NO_2$".

Accordingly, we plan to change the title of our companion manuscript to:

"Multiphase Reaction of $SO_2$ on $CaCO_3$ Particles. 2. $NO_2$-initiated Oxidation of $SO_2$ by $O_2$".

*Specific comments*

*Introduction: (1) P.3, lines 59-62: It would be worth to mention also the studies from 2001 (Turšič et al. 2001, Atmos. Environ.).*

**Response:** Accepted.

In the revised manuscript, we have added Turšič et al. (2001) in our citation.

*Experimental: (2) The experimental approach (Raman mapping analysis) where you can follow the changes during the reaction course (as can be seen in Fig. S2) is intriguing.*

**Response:** We thank the reviewer for the supporting remark.

*Results and discussion: (3) Fig. 2 nicely shows how the peak for $CO_3^{2-}$ decreases and disappears after certain time of reaction; the change can be seen also in Fig. 3. What happens to it (releases as $CO_2$)? (4) It is not correct to explain its "disappearance" as "completely consumed" (line 171).*

**Response:**

$CO_3^{2-}$ was converted to $CO_2$ by the reaction with $H^+$, which was released into the gas phase. In the revised manuscript, we have briefly discussed this process.

"The formation of $Ca(NO_3)_2$ started with the reaction of $NO_2$ with adsorbed or liquid water, forming $HNO_3$ and $HNO_2$. Then $HNO_3$ reacted with $CaCO_3$ forming $Ca(NO_3)_2$ as well as $CO_2$, which was released to the gas phase."

In the revised manuscript, we have rephrased the "completely consumed" to "completely reacted".

*(5) Line 186: "the consumption of CaCO₃" is not appropriate*

**Response:** Accepted.

In the revised manuscript, we have changed it to "the decline of $CaCO_3$".

*(6) Lines 186-187: If Ca(NO₃)₂ is formed in the reaction between CaCO₃ and NO₂, NO₂ should first disproportionate to NO₃⁻ and NO₂⁻, which is possible in the presence of water. How is then Ca(NO₃)₂ first formed from CaCO₃, and only then converts into droplet in the presence of water? The authors should explain the reactions also for the first step, i.e. the conversion of CaCO₃ to Ca(NO₃)₂ although the reference is given (line 188). I suggest that the complete mechanism is written.*

**Response:**

The details of the mechanism of the reaction of $CaCO_3$ with $NO_2$ are reported in our previous paper (Li et al., 2010). In the revised manuscript, we have added the following texts and reaction equations:

$Ca(NO_3)_2$ has been observed in the reaction of $CaCO_3$ with $NO_2$ in previous studies (Li et al., 2010; Tan et al., 2017). The formation of $Ca(NO_3)_2$ started with the reaction of $NO_2$ with adsorbed or liquid water, forming $HNO_3$ and $HNO_2$. Then $HNO_3$ reacted with $CaCO_3$ forming $Ca(NO_3)_2$ as well as $CO_2$, which was released to the gas phase. The reaction equations are as follows:

$$NO_2(g) \leftrightarrow NO_2(aq) \tag{R1}$$

$$2NO_2(aq) + H_2O(aq) \longrightarrow HNO_3(aq) + HNO_2(aq) \tag{R2}$$

$$HNO_3(aq) \longrightarrow H^+(aq) + NO_3^-(aq) \tag{R3}$$

$$CaCO_3(s) + H^+(aq) \longrightarrow Ca^+(aq) + HCO_3^-(aq) \tag{R4}$$

$$HCO_3^-(aq) + H^+(aq) \longrightarrow H_2O(aq) + CO_2(g) \tag{R5}$$

*(7) The authors may want to add a reference of Tan et al., 2016, ACP.*

**Response:** Accepted.

In the revised manuscript, we have added Tan et al. (2016) as a reference.

*(8) It is concluded that aqueous phase plays a key role in $SO_2$ oxidation by $NO_2$, which is also known from previous studies. Line 219: pH is estimated to be around 3. What would be the concentrations of reactive species in $Ca(NO_3)_2$ droplet?*

**Response:**

We suppose that the reviewer referred to the concentrations of S(IV) species. The concentrations of $HSO_3^-$, $H_2SO_3$, and $SO_3^{2-}$ were estimated to be $\sim1.1\times10^{-3}$, $9.2\times10^{-5}$, and $6.6\times10^{-8}$ mol $L^{-1}$, respectively, using the equilibrium constants in Seinfeld and Pandis (2006).

We have added these values in the revised manuscript.

"The concentrations of $HSO_3^-$, $H_2SO_3$, and $SO_3^{2-}$ were estimate to be $\sim1.1\times10^{-3}$, $9.2\times10^{-5}$, and $6.6\times10^{-8}$ mol $L^{-1}$, respectively, using the equilibrium constants in Seinfeld and Pandis (2006) and thus the main S(IV) species was $HSO_3^-$."

*(9) Lines 236-241: This part is not well understandable. It is concluded that precipitation of $CaSO_4$ formed in/on $Ca(NO_3)_2$ droplet promotes sulfate formation. On the other hand, when $NaNO_3$ or $NH_4NO_3$ droplet is used instead of $Ca(NO_3)_2$, no sulfate was formed after 300 min. If aqueous phase is a key factor for the oxidation of $SO_2$ with $NO_2$, then this should happen also in these droplets, although the reaction is most probably much slower. Why the reaction was not carried out at longer times?*

**Response:**

The purpose of the comparison between the reaction of $NaNO_3$ and $NH_4NO_3$ and the reaction of $Ca(NO_3)_2$ was to qualitatively examine the effect of cations on sulfate formation rate. At 300 min, sulfate was readily detectable in the reaction of $Ca(NO_3)_2$ while it was below the detection limit in the reaction of $NaNO_3$ and $NH_4NO_3$ (Fig. 5, Fig. 6). Although sulfate may have been formed, the absence of sulfate at 300 min shows that the sulfate production was extremely slow. The difference by 300 min has clearly indicated that the sulfate formation in the reaction of $Ca(NO_3)_2$ was much faster than that in the reaction the $NaNO_3$ and $NH_4NO_3$ and $Ca^{2+}$ promoted sulfate formation, which likely resulted from $CaSO_4$ precipitation. Therefore, the reaction was not continued for longer times.

In the revised manuscript, we have made some changes to improve the clarity of the discussion. Now it reads:

"Based on Raman spectra, we found that in the reaction of a $NaNO_3$ or a $NH_4NO_3$ droplet with $NO_2/SO_2$ sulfate was below the detection limit after 300 min in the same reaction conditions as $Ca(NO_3)_2$ and $CaCO_3$ (Fig. 6 and Table 1). Accordingly, no sulfate solid particles were observed in these droplets. Clearly, the sulfate production rate was larger in the presence of $Ca^{2+}$ compared to those in the presence of $Na^+$ or $NH_4^+$. The difference can be explained by the change of Gibbs energy."

"$\Delta_r G$ increases with increasing sulfate concentration. According to the different results between the reaction on $Ca(NO_3)_2$ droplet and the reaction on $NaNO_3$ and $NH_4NO_3$ droplet, there might be a backward reaction of $SO_2$ oxidation which consumed sulfate, although the detailed mechanism of the backward reaction is unknown at the moment. For $NaNO_3$ and $NH_4NO_3$ droplet, once sulfate concentration reached certain level, the reaction may stop due to the increase of $\Delta_r G$. For $Ca(NO_3)_2$ droplet, the precipitation of $CaSO_4$ can substantially decrease the activity of $SO_4^{2-}$, and thus decrease $\Delta_r G$ and promote the oxidation of $SO_2$ and sulfate formation."

*(10) Line 240: In droplets of NaNO₃ or NH₄NO₃, CaSO₄ cannot be formed.*

**Response:**

Accepted. In the revised manuscript, we have changed "$CaSO_4$" to "sulfate".

*(11) Line 250: Is it correct that at RH of 46% the conditions for a complete conversion into a Ca(NO₃)₂ droplet are achieved?*

**Response:**

Yes. We observed that a complete conversion from $CaCO_3$ particle to $Ca(NO_3)_2$ droplet occurred at 46% RH and then sulfate was observed.

*(12) Line 259: Write what is DRIFTS technique (it was not mentioned before).*

**Response:**

Accepted. In the revised manuscript, we have provided the full name of DRIFTS as "Diffuse Reflectance Infrared Fourier Transform Spectroscopy".

*(13) Line 206: ATD particles?*

**Response:**

In the revised manuscript, we have provided the full name of ATD as "Arizona Test Dust".

*(14) Lines 273-275: Is this statement correct? Higher concentrations of aqueous sulfate may suppress the reaction between $SO_2$ and $NO_2$, while $CaSO_4$ precipitation can promote it.*

**Response:**

As we found on the effect of cations (Section 3.3.2), reduced sulfate concentration by $CaSO_4$ precipitation likely led to the enhanced sulfate production rate in the reaction of $SO_2$ on $Ca(NO_3)_2$. According to Eq. 5, higher sulfate concentration could increase the reaction Gibbs energy $\Delta_rG$ and thus suppress the reaction of $SO_2$ and $NO_2$.

In the revised manuscript, we have further explained this statement.

"According to the effect of cations (Section 3.3.2), while reduced sulfate concentration by $CaSO_4$ precipitation likely led to the enhanced sulfate production rate in the reaction of $SO_2$ on $Ca(NO_3)_2$, higher sulfate concentration could increase the reaction Gibbs energy $\Delta_rG$ (as shown in Eq. 5) and thus suppress the reaction of $SO_2$ and $NO_2$. This can reduce the uptake coefficient of $SO_2$."

**References**

Cheng, Y. F., Zheng, G. J., Wei, C., Mu, Q., Zheng, B., Wang, Z. B., Gao, M., Zhang, Q., He, K. B., Carmichael, G., Poschl, U., and Su, H.: Reactive nitrogen chemistry in aerosol water as a source of sulfate during haze events in China, Sci. Adv., 2, 10.1126/sciadv.1601530, 2016.

Li, H. J., Zhu, T., Zhao, D. F., Zhang, Z. F., and Chen, Z. M.: Kinetics and mechanisms of heterogeneous reaction of $NO_2$ on $CaCO_3$ surfaces under dry and wet conditions, Atmos. Chem. Phys., 10, 463-474, 2010.

Seinfeld, J. H., and Pandis, S. N.: Atmospheric chemistry and physics: from air pollution to climate change, 2nd ed., John Wiley &Sons. Inc., 2006.

Tan, F., Jing, B., Tong, S. R., and Ge, M. F.: The effects of coexisting Na2SO4 on heterogeneous uptake of NO2 on CaCO3 particles at various RHs, Sci. Total Environ. , 586, 930-938, 10.1016/j.scitotenv.2017.02.072, 2017.

Wang, G., Zhang, R., Gomez, M. E., Yang, L., Levy Zamora, M., Hu, M., Lin, Y., Peng, J., Guo, S., Meng, J., Li, J., Cheng, C., Hu, T., Ren, Y., Wang, Y., Gao, J., Cao, J., An, Z., Zhou, W., Li, G., Wang, J., Tian, P., Marrero-Ortiz, W., Secrest, J., Du, Z., Zheng, J., Shang, D., Zeng, L., Shao, M., Wang, W., Huang, Y., Wang, Y., Zhu, Y., Li, Y., Hu, J., Pan, B., Cai, L., Cheng, Y., Ji, Y., Zhang, F., Rosenfeld, D., Liss, P. S., Duce, R. A., Kolb, C. E., and Molina, M. J.: Persistent sulfate formation from London Fog to Chinese haze, Proc. Nat. Acad. Sci. U.S.A., 113, 13630-13635, 10.1073/pnas.1616540113, 2016.

Xue, J., Yuan, Z. B., Griffith, S. M., Yu, X., Lau, A. K. H., and Yu, J. Z.: Sulfate Formation Enhanced by a Cocktail of High NOx, SO2, Particulate Matter, and Droplet pH during Haze-Fog Events in Megacities in China: An Observation-Based Modeling Investigation, Environ. Sci. Technol., 50, 7325-7334, 10.1021/acs.est.6b00768, 2016.

Yu, T., Zhao, D., Song, X., and Zhu, T.: Multiphase Reaction of $SO_2$ with $NO_2$ on $CaCO_3$ Particles. 2. $NO_2$-initialized Oxidation of $SO_2$ by $O_2$, Atmos. Chem. Phys. Discuss., 2017, 1-20, 10.5194/acp-2017-900, 2017.

---

## Author Comment (AC2) · 10 Oct 2017

**Responses to Referee # 2**

We thank the reviewer for carefully reviewing our manuscript. The comments and suggestions are greatly appreciated. All the comments have been addressed. In the following, please find our responses to the comments one by one and corresponding revisions made to the manuscript. The original comments are shown in italics. The revised parts of the manuscript are highlighted.

*This study investigated the heterogeneous reaction of $SO_2$ with $NO_2$ on individual $CaCO_3$ particles in $N_2$ using Micro-Raman spectroscopy. The results show that $CaCO_3$ was first converted to $Ca(NO_3)_2$ forming a droplet and promoting the oxidation of $SO_2$ by $NO_2$. The precipitation of $CaSO_4$ was suggested as a key step accelerating the sulfate formation. Based on the uptake coefficient determined, the authors concluded that the $SO_2 + NO_2$ reaction was not important compared to the oxidation of $SO_2$ by OH radicals. The experiment was well designed and the paper was well written.*

*But I do have concerns about the role of $CaSO_4$ precipitation and I would also suggest the authors to compare their results with literature data before making strong statement on the role of $NO_2+SO_2$ chemistry.*

*Major concern:*

*1. The authors generalized the results of their $CaCO_3$ experiments to assess the role of $NO_2+SO_2$ chemistry. I am not sure if such generalization is correct because according to early studies of Lee and Schwartz,1983 and Clifton et al.,1988, this reaction can be important under polluted and less acidic conditions in contrary to the authors' statement. The authors used deposited super-micro particles in their experiments. But I don't expect much difference between such a system and bulk experiments because large particles are not subject to strong Kelvin effect and particles contacted with substrates would not become supersaturated solution of high ionic strength due to nucleation. Thus before generalizing results for ambient aerosols, I would suggest the authors to discuss their difference with those early studies.*

**Response:**

We thank the reviewer for this comment and suggestion.

In the revised manuscript, we have added the following paragraph to discuss the comparison of our study with previous studies using bulk solution.

"The $\gamma$ of $SO_2$ was further compared with the reaction rate constants of the aqueous reaction of $NO_2$ with sulfite and bisulfite in bulk solution in the literature by deriving $\gamma$ from rate constants using the method in Davidovits et al. (2006). The detailed method can be referred to the supplement S1. Lee and Schwartz (1983) determined the rate constant of the reaction of $NO_2$ with bisulfite to be $>2\times10^6$

mol$^{-1}$ L s$^{-1}$ at pH 5.8 and 6.4. Clifton et al. (1988) determined the rate constant of the reaction of NO$_2$ with sulfite/bisulfite to be (1.24-2.95)$\times 10^7$ mol$^{-1}$ L s$^{-1}$ at pH 5.6-13 and further reported a rate constant of 1.4$\times 10^5$ mol$^{-1}$ L s$^{-1}$ at pH 5 from the study of Lee and Schwartz (1983). The different rate constants were attributed to the different approaches to determine the reaction rate by Clifton et al. (1988). Clifton et al. (1988) determined the reaction rate from the consumption rate the reactant, NO$_2$, which corresponds to the first reaction step of NO$_2$ with S(IV). Yet, Lee and Schwartz (1983) determined the reaction rate from the production rate of products (their conductivity), which is expected to be much slower than NO$_2$ consumption since formation of products needs more steps. In this study, we determined $\gamma$ using sulfate production rate, and thus our data are comparable to the study of Lee and Schwartz (1983). Yet, the study of Lee and Schwartz (1983) only covers a pH range of 5-6.4 and has no overlap with the pH (~3) in our study, therefore uptake coefficients from both studies are not directly comparable. Nevertheless, the reaction rate of 1.4$\times 10^5$ mol$^{-1}$ L s$^{-1}$ at pH 5 corresponds to the uptake coefficient of 4.3$\times 10^{-7}$, which is around one order of magnitude higher than the uptake coefficient in our study determined at pH ~3 for the droplet. The difference may be due to the different pH between these two studies, the different mechanisms between the multiphase reaction on particles and bulk aqueous reaction, and the different concentrations of each S(IV) species since the different species may have different reactivity with NO$_2$. The reaction rate of S(IV) has been found to decrease with decreasing pH and the reactivity of sulfite with NO$_2$ seems to be higher than bisulfite (Lee and Schwartz, 1983; Clifton et al., 1988; Takeuchi et al., 1977). In addition, the ionic strength in the droplet of this study (15-55 mol Kg$^{-1}$) was much higher than that in the bulk solution in previous studies (on the order of 10$^{-6}$-10$^{-1}$ mol Kg$^{-1}$), which may also influence the reaction rate."

*2. Based on Equation (5), the authors concluded that the precipitation-induced reduction of sulfate will promote the oxidation of SO$_2$ by NO$_2$ (reaction 2). I don't know if it is correct to use Eq. (5) in this way. Because Equation (5) is valid for reversible reactions and removing/adding products of non-reversible reactions will not change the reaction rate much.*

**Response:**

Using the change of Gibbs energy to express the spontaneity of a reaction is applicable to all reactions. Moreover, in theory, all chemical reactions are reversible, to some extent (Keeler and Wothers, 2008; de Nevers, 2012). According to the different results between the reaction on Ca(NO$_3$)$_2$ droplet and the reaction on NaNO$_3$ and NH$_4$NO$_3$ droplet, there might be a backward reaction of SO$_2$ oxidation which consumed sulfate, although the detailed mechanism of the backward reaction is unknown at the moment. Therefore, we used Equation (5) to explain the difference between the reaction on Ca(NO$_3$)$_2$ droplet and the reaction on NaNO$_3$ and NH$_4$NO$_3$ droplet.

In the revised manuscript, we have modified the discussion:

"$\Delta_r G$ increases with increasing sulfate concentration. According to the different results between the reaction on $Ca(NO_3)_2$ droplet and the reaction on $NaNO_3$ and $NH_4NO_3$ droplet, there might be a backward reaction of $SO_2$ oxidation which consumed sulfate, although the detailed mechanism of the backward reaction is unknown at the moment. For $NaNO_3$ and $NH_4NO_3$ droplet, once sulfate concentration reached certain level, the reaction may stop due to the increase of $\Delta_r G$. For $Ca(NO_3)_2$ droplet, the precipitation of $CaSO_4$ can substantially decrease the activity of $SO_4^{2-}$, and thus decrease $\Delta_r G$ and promote the oxidation of $SO_2$ and sulfate formation."

*Other comments:*

*Page 5 line 133, half sentene?*

**Response:**

Accepted. In the revised manuscript, we have fixed this sentence. Now it reads:

"In order to minimize the influence variations of incident laser on Raman intensity, these seven particles were measured before each experiment..."

*Page 6 line 187, I would suggest to briefly describe the mechanism of $Ca(NO_3)_2$ formation.*

**Response:**

Accepted. In the revised, we have briefly described the reaction mechanism of $Ca(NO_3)_2$ formation and provided a brief mechanism of the reaction of $CaCO_3$ with $NO_2$. Now it reads:

"$Ca(NO_3)_2$ has been observed in the reaction of $CaCO_3$ with $NO_2$ in previous studies (Li et al., 2010; Tan et al., 2016). The formation of $Ca(NO_3)_2$ started with the reaction of $NO_2$ with adsorbed or liquid water, forming $HNO_3$ and $HNO_2$. Then $HNO_3$ reacted with $CaCO_3$ forming $Ca(NO_3)_2$ as well as $CO_2$, which was released to the gas phase. The reaction equations are as follows:

$$NO_2(g) \leftrightarrow NO_2(aq) \tag{R1}$$

$$2NO_2(aq) + H_2O(aq) \rightarrow HNO_3(aq) + HNO_2(aq) \tag{R2}$$

$$HNO_3(aq) \rightarrow H^+(aq) + NO_3^-(aq) \tag{R3}$$

$$CaCO_3(s) + H^+(aq) \rightarrow Ca^+(aq) + HCO_3^-(aq) \tag{R4}$$

$$HCO_3^-(aq) + H^+(aq) \rightarrow H_2O(aq) + CO_2(g) \tag{R5}$$"

*Will the present of $SO_2$ influence the uptake of $NO_2$?*

**Response:**

In principle, the presence of $SO_2$ should enhance $NO_2$ uptake due to its reaction with $NO_2$. However, as we observed in our study, the reactive uptake of $NO_2$ on $CaCO_3$ particles was much faster than the

reaction of $NO_2$ with $SO_2$ and sulfate as the reaction product of $SO_2$ was essentially only observed after $CaCO_3$ was completely converted to $Ca(NO_3)_2$ droplet by $NO_2$. Therefore, the influence of $SO_2$ on $NO_2$ uptake was not significant.

In the revised manuscript, we have added one sentence to discuss this point.

"The much faster $Ca(NO_3)_2$ formation due to the $NO_2$ uptake on $CaCO_3$ particle compared with the reaction of $SO_2$ with $NO_2$ and sulfate appearing only after the complete conversion of $CaCO_3$ indicate that the reaction of $SO_2$ with $NO_2$ does not contributed significantly to $NO_2$ uptake."

*Fig. 4, no data for nitrate and carbonate after 120 min, why?*

**Response:**

By 118 min, $CaCO_3$ was completely converted to $Ca(NO_3)_2$. Carbonate had decreased to zero and nitrate had reached a plateau. Therefore no further data of carbonate and nitrate were shown. In the revised manuscript, we have explained this in the caption of Fig. 4.

**References**

Clifton, C. L., Altstein, N., and Huie, R. E.: Rate-constant for the reaction of $NO_2$ with sulfur(IV) over the pH range 5.3-13, Environ. Sci. Technol., 22, 586-589, 10.1021/es00170a018, 1988.

Davidovits, P., Kolb, C. E., Williams, L. R., Jayne, J. T., and Worsnop, D. R.: Mass accommodation and chemical reactions at gas-liquid interfaces, Chem. Rev., 106, 1323-1354, 10.1021/cr040366k, 2006.

de Nevers, N.: Physical and Chemical Equilibrium for Chemical Engineers, 2nd ed., John Wiley & Sons, Inc., Hoboken, New Jersey, 348 pp., 2012.

Keeler, J., and Wothers, P.: Chemical Structure and Reactivity: An Integrated Approach, 1st ed., Oxford University Press, Oxford, 924 pp., 2008.

Lee, Y.-N., and Schwartz, S. E.: Kinetics of oxidation of aqueous sulfur (IV) by nitrogen dioxide, in: Precipitation Scavenging, Dry Deposition and Resuspension, edited by: Pruppacher, H. R., Semonin, R. G., and Slinn, W. G. N., Elsevier, New York, 453-466, 1983.

Li, H. J., Zhu, T., Zhao, D. F., Zhang, Z. F., and Chen, Z. M.: Kinetics and mechanisms of heterogeneous reaction of $NO_2$ on $CaCO_3$ surfaces under dry and wet conditions, Atmos. Chem. Phys., 10, 463-474, 2010.

Takeuchi, H., Ando, M., and Kizawa, N.: Absorption of Nitrogen Oxides in Aqueous Sodium Sulfite and Bisulfite Solutions, Industrial & Engineering Chemistry Process Design and Development, 16, 303-308, 10.1021/i260063a010, 1977.

Tan, F., Tong, S. R., Jing, B., Hou, S. Q., Liu, Q. F., Li, K., Zhang, Y., and Ge, M. F.: Heterogeneous reactions of $NO_2$ with $CaCO_3$-$(NH_4)(2)SO_4$ mixtures at different relative humidities, Atmos. Chem. Phys., 16, 8081-8093, 10.5194/acp-16-8081-2016, 2016.

---

## Author Response (AR1)

**Responses to Referee # 1**

We thank the reviewer for carefully reviewing our manuscript. The comments and suggestions are greatly appreciated. All the comments have been addressed. In the following, please find our responses to the comments one by one and corresponding revisions made to the manuscript. The original comments are shown in italics. The revised parts of the manuscript are highlighted.

*General comments*

*The paper presents the results on multiphase reaction of $SO_2$ with $NO_2$ on individual $CaCO_3$ particles in $N_2$ at RH between 17 and 72% using Micro-Raman spectrometer with a flow reaction system. The reaction process was systematically investigated and found that $CaCO_3$ converts first to $Ca(NO_3)_2$ (by the reaction with $NO_2$) and its deliquescence to droplet, where further $SO_2$ oxidizes with $NO_2$ forming $CaSO_4$. The reactive uptake coefficient $\gamma$ of $SO_2$ was determined on the basis of sulfate formation rate.*

*Although many studies concerning $SO_2$ oxidation in the atmosphere were performed in late 1980s and 1990s, mainly due to much higher pollution with $SO_2$ all over the world at that time, and various pathways of oxidation were identified, the questions concerning sulfate formation have not yet been fully resolved. As shown e.g., that high concentrations of sulfate during heavily polluted episodes under haze conditions in China could not be explained on the basis of known pathways only. In addition, due to the progress and development of techniques, nowadays there are more possibilities to study processes also on the micro level as shown in this paper.*

*Thus, I found the paper of sufficient atmospheric interest to merit publication after revision; in "specific comments" some questions and/or comments are listed which should be considered.*

*However, I would strongly recommend showing also the results for the system $SO_2/NO_2/H_2O/O_2$ together with those presented here and not in the next paper as said in line 80. Experimental conditions will be closer to atmospheric, and as mentioned on p.9 (lines 301-303) it is expected that the reactions under O2 are faster and could be more important source of sulfate.*

**Response:**

We thank the reviewer for the supporting remarks.

As to the recommendation "*showing also the results for the system $SO_2/NO_2/H_2O/O_2$ together with those presented here and not in the next paper as said in line 80*", we realized that our phrasing in the some texts of the manuscript was not precise and clear enough to express our primary motivation. We have modified these texts (e.g. lines 80-83, lines 350-357) in the revised manuscript to clearly state our motivation, i.e., to address the multiphase reaction of $SO_2$ directly with $NO_2$ and evaluate the importance of this reaction pathway in sulfate formation in the real atmosphere, which was proposed by a number of recent studies (Cheng et al., 2016; Wang et al., 2016; Xue et al., 2016) but remains unclear. In order to exclude potential confounding reactions from other compounds, we used inert $N_2$ as a carrier gas. For this motivation, the direct reaction of $SO_2$ with $NO_2$ has ambient relevance no matter whether $O_2$ is present or not.

The reason that we did not include the multiphase reaction of $SO_2$ with $O_2/NO_2$ in this paper is because we found that in this reaction $SO_2$ was actually oxidized by $O_2$, not by $NO_2$. It is distinct from the reaction of $SO_2$ directly with $NO_2$ with markedly different mechanisms, products, and atmospheric implications, as we will show (Yu et al., 2017). $O_2$ was the main oxidant of $SO_2$ and $NO_2$ only acted as an initiator of chain reactions. The atmospheric implications are significantly different from the direct reaction of $SO_2$ with $NO_2$ because not only the oxidation of $SO_2$ by $O_2$ leads to much faster sulfate oxidation but also the reaction is not linked to reactive nitrogen chemistry in the atmosphere. Therefore, we address this reaction in a separate companion manuscript (Yu et al., 2017).

In addition, in order to reflect the distinction of these two studies more precisely, we have revised the title of our manuscript as follows:

"Multiphase Reaction of $SO_2$ on $CaCO_3$ Particles. 1. Oxidation of $SO_2$ by $NO_2$".

Accordingly, we plan to change the title of our companion manuscript to:

"Multiphase Reaction of $SO_2$ on $CaCO_3$ Particles. 2. $NO_2$-initiated Oxidation of $SO_2$ by $O_2$".

*Specific comments*

*Introduction: (1) P.3, lines 59-62: It would be worth to mention also the studies from 2001 (Turšič et al. 2001, Atmos. Environ.).*

**Response:** Accepted.

In the revised manuscript, we have added Turšič et al. (2001) in our citation.

*Experimental: (2) The experimental approach (Raman mapping analysis) where you can follow the changes during the reaction course (as can be seen in Fig. S2) is intriguing.*

**Response:** We thank the reviewer for the supporting remark.

*Results and discussion: (3) Fig. 2 nicely shows how the peak for $CO_3^{2-}$ decreases and disappears after certain time of reaction; the change can be seen also in Fig. 3. What happens to it (releases as $CO_2$)? (4) It is not correct to explain its "disappearance" as "completely consumed" (line 171).*

**Response:**

$CO_3^{2-}$ was converted to $CO_2$ by the reaction with $H^+$, which was released into the gas phase. In the revised manuscript, we have briefly discussed this process.

"The formation of $Ca(NO_3)_2$ started with the reaction of $NO_2$ with adsorbed or liquid water, forming $HNO_3$ and $HNO_2$. Then $HNO_3$ reacted with $CaCO_3$ forming $Ca(NO_3)_2$ as well as $CO_2$, which was released to the gas phase."

In the revised manuscript, we have rephrased the "completely consumed" to "completely reacted".

*(5) Line 186: "the consumption of CaCO₃" is not appropriate*

**Response:** Accepted.

In the revised manuscript, we have changed it to "the decline of $CaCO_3$".

*(6) Lines 186-187: If Ca(NO₃)₂ is formed in the reaction between CaCO₃ and NO₂, NO₂ should first disproportionate to NO₃⁻ and NO₂⁻, which is possible in the presence of water. How is then Ca(NO₃)₂ first formed from CaCO₃, and only then converts into droplet in the presence of water? The authors should explain the reactions also for the first step, i.e. the conversion of CaCO₃ to Ca(NO₃)₂ although the reference is given (line 188). I suggest that the complete mechanism is written.*

**Response:**

The details of the mechanism of the reaction of $CaCO_3$ with $NO_2$ are reported in our previous paper (Li et al., 2010). In the revised manuscript, we have added the following texts and reaction equations:

$Ca(NO_3)_2$ has been observed in the reaction of $CaCO_3$ with $NO_2$ in previous studies (Li et al., 2010; Tan et al., 2017). The formation of $Ca(NO_3)_2$ started with the reaction of $NO_2$ with adsorbed or liquid water, forming $HNO_3$ and $HNO_2$. Then $HNO_3$ reacted with $CaCO_3$ forming $Ca(NO_3)_2$ as well as $CO_2$, which was released to the gas phase. The reaction equations are as follows:

$$NO_2(g) \leftrightarrow NO_2(aq) \tag{R1}$$

$$2NO_2(aq) + H_2O(aq) \longrightarrow HNO_3(aq) + HNO_2(aq) \tag{R2}$$

$$HNO_3(aq) \longrightarrow H^+(aq) + NO_3^-(aq) \tag{R3}$$

$$CaCO_3(s) + H^+(aq) \longrightarrow Ca^+(aq) + HCO_3^-(aq) \tag{R4}$$

$$HCO_3^-(aq) + H^+(aq) \longrightarrow H_2O(aq) + CO_2(g) \tag{R5}$$

*(7) The authors may want to add a reference of Tan et al., 2016, ACP.*

**Response:** Accepted.

In the revised manuscript, we have added Tan et al. (2016) as a reference.

*(8) It is concluded that aqueous phase plays a key role in SO₂ oxidation by NO₂, which is also known from previous studies. Line 219: pH is estimated to be around 3. What would be the concentrations of reactive species in Ca(NO₃)₂ droplet?*

**Response:**

We suppose that the reviewer referred to the concentrations of S(IV) species. The concentrations of $HSO_3^-$, $H_2SO_3$, and $SO_3^{2-}$ were estimated to be $\sim 1.1\times10^{-3}$, $9.2\times10^{-5}$, and $6.6\times10^{-8}$ mol $L^{-1}$, respectively, using the equilibrium constants in Seinfeld and Pandis (2006).

We have added these values in the revised manuscript.

"The concentrations of $HSO_3^-$, $H_2SO_3$, and $SO_3^{2-}$ were estimate to be $\sim 1.1\times10^{-3}$, $9.2\times10^{-5}$, and $6.6\times10^{-8}$ mol $L^{-1}$, respectively, using the equilibrium constants in Seinfeld and Pandis (2006) and thus the main S(IV) species was $HSO_3^-$."

*(9) Lines 236-241: This part is not well understandable. It is concluded that precipitation of CaSO₄ formed in/on Ca(NO₃)₂ droplet promotes sulfate formation. On the other hand, when NaNO₃ or NH₄NO₃ droplet is used instead of Ca(NO₃)₂, no sulfate was formed after 300 min. If aqueous phase is a key factor for the oxidation of SO₂ with NO₂, then this should happen also in these droplets, although the reaction is most probably much slower. Why the reaction was not carried out at longer times?*

**Response:**

The purpose of the comparison between the reaction of NaNO₃ and NH₄NO₃ and the reaction of Ca(NO₃)₂ was to qualitatively examine the effect of cations on sulfate formation rate. At 300 min, sulfate was readily detectable in the reaction of Ca(NO₃)₂ while it was below the detection limit in the reaction of NaNO₃ and NH₄NO₃ (Fig. 5, Fig. 6). Although sulfate may have been formed, the absence of sulfate at 300 min shows that the sulfate production was extremely slow. The difference by 300 min has clearly indicated that the sulfate formation in the reaction of Ca(NO₃)₂ was much faster than that in the reaction the NaNO₃ and NH₄NO₃ and $Ca^{2+}$ promoted sulfate formation, which likely resulted from CaSO₄ precipitation. Therefore, the reaction was not continued for longer times.

In the revised manuscript, we have made some changes to improve the clarity of the discussion. Now it reads:

"Based on Raman spectra, we found that in the reaction of a $NaNO_3$ or a $NH_4NO_3$ droplet with $NO_2/SO_2$ sulfate was below the detection limit after 300 min in the same reaction conditions as $Ca(NO_3)_2$ and $CaCO_3$ (Fig. 6 and Table 1). Accordingly, no sulfate solid particles were observed in these droplets. Clearly, the sulfate production rate was larger in the presence of $Ca^{2+}$ compared to those in the presence of $Na^+$ or $NH_4^+$. The difference can be explained by the change of Gibbs energy."

"$\Delta_rG$ increases with increasing sulfate concentration. According to the different results between the reaction on $Ca(NO_3)_2$ droplet and the reaction on $NaNO_3$ and $NH_4NO_3$ droplet, there might be a backward reaction of $SO_2$ oxidation which consumed sulfate, although the detailed mechanism of the backward reaction is unknown at the moment. For $NaNO_3$ and $NH_4NO_3$ droplet, once sulfate concentration reached certain level, the reaction may stop due to the increase of $\Delta_rG$. For $Ca(NO_3)_2$ droplet, the precipitation of $CaSO_4$ can substantially decrease the activity of $SO_4^{2-}$, and thus decrease $\Delta_rG$ and promote the oxidation of $SO_2$ and sulfate formation."

*(10) Line 240: In droplets of NaNO₃ or NH₄NO₃, CaSO₄ cannot be formed.*

**Response:**

Accepted. In the revised manuscript, we have changed "$CaSO_4$" to "sulfate".

*(11) Line 250: Is it correct that at RH of 46% the conditions for a complete conversion into a Ca(NO₃)₂ droplet are achieved?*

**Response:**

Yes. We observed that a complete conversion from $CaCO_3$ particle to $Ca(NO_3)_2$ droplet occurred at 46% RH and then sulfate was observed.

*(12) Line 259: Write what is DRIFTS technique (it was not mentioned before).*

**Response:**

Accepted. In the revised manuscript, we have provided the full name of DRIFTS as "Diffuse Reflectance Infrared Fourier Transform Spectroscopy".

*(13) Line 206: ATD particles?*

**Response:**

In the revised manuscript, we have provided the full name of ATD as "Arizona Test Dust".

*(14) Lines 273-275: Is this statement correct? Higher concentrations of aqueous sulfate may suppress the reaction between $SO_2$ and $NO_2$, while $CaSO_4$ precipitation can promote it.*

**Response:**

As we found on the effect of cations (Section 3.3.2), reduced sulfate concentration by $CaSO_4$ precipitation likely led to the enhanced sulfate production rate in the reaction of $SO_2$ on $Ca(NO_3)_2$. According to Eq. 5, higher sulfate concentration could increase the reaction Gibbs energy $\Delta_r G$ and thus suppress the reaction of $SO_2$ and $NO_2$.

In the revised manuscript, we have further explained this statement.

"According to the effect of cations (Section 3.3.2), while reduced sulfate concentration by $CaSO_4$ precipitation likely led to the enhanced sulfate production rate in the reaction of $SO_2$ on $Ca(NO_3)_2$, higher sulfate concentration could increase the reaction Gibbs energy $\Delta_r G$ (as shown in Eq. 5) and thus suppress the reaction of $SO_2$ and $NO_2$. This can reduce the uptake coefficient of $SO_2$."

**Responses to Referee # 2**

We thank the reviewer for carefully reviewing our manuscript. The comments and suggestions are greatly appreciated. All the comments have been addressed. In the following, please find our responses to the comments one by one and corresponding revisions made to the manuscript. The original comments are shown in italics. The revised parts of the manuscript are highlighted.

*This study investigated the heterogeneous reaction of $SO_2$ with $NO_2$ on individual $CaCO_3$ particles in $N_2$ using Micro-Raman spectroscopy. The results show that $CaCO_3$ was first converted to $Ca(NO_3)_2$ forming a droplet and promoting the oxidation of $SO_2$ by $NO_2$. The precipitation of $CaSO_4$ was suggested as a key step accelerating the sulfate formation. Based on the uptake coefficient determined, the authors concluded that the $SO_2 + NO_2$ reaction was not important compared to the oxidation of $SO_2$ by OH radicals. The experiment was well designed and the paper was well written.*

*But I do have concerns about the role of $CaSO_4$ precipitation and I would also suggest the authors to compare their results with literature data before making strong statement on the role of $NO_2+SO_2$ chemistry.*

*Major concern:*

*1. The authors generalized the results of their $CaCO_3$ experiments to assess the role of $NO_2+SO_2$ chemistry. I am not sure if such generalization is correct because according to early studies of Lee and Schwartz,1983 and Clifton et al.,1988, this reaction can be important under polluted and less acidic conditions in contrary to the authors' statement. The authors used deposited super-micro particles in their experiments. But I don't expect much difference between such a system and bulk experiments because large particles are not subject to strong Kelvin effect and particles contacted with substrates would not become supersaturated solution of high ionic strength due to nucleation. Thus before generalizing results for ambient aerosols, I would suggest the authors to discuss their difference with those early studies.*

**Response:**

We thank the reviewer for this comment and suggestion.

In the revised manuscript, we have added the following paragraph to discuss the comparison of our study with previous studies using bulk solution.

==“The $\gamma$ of $SO_2$ was further compared with the reaction rate constants of the aqueous reaction of $NO_2$ with sulfite and bisulfite in bulk solution in the literature by deriving $\gamma$ from rate constants using the method in Davidovits et al. (2006). The detailed method can be referred to the supplement S1. Lee and Schwartz (1983) determined the rate constant of the reaction of $NO_2$ with bisulfite to be >2×10$^6$==

mol$^{-1}$ L$^{-1}$ s$^{-1}$ at pH 5.8 and 6.4. Clifton et al. (1988) determined the rate constant of the reaction of NO$_2$ with sulfite/bisulfite to be (1.24-2.95)×10$^7$ mol$^{-1}$ L$^{-1}$ s$^{-1}$ at pH 5.6-13 and further reported a rate constant of 1.4×10$^5$ mol$^{-1}$ L$^{-1}$ s$^{-1}$ at pH 5 from the study of Lee and Schwartz (1983). The different rate constants were attributed to the different approaches to determine the reaction rate by Clifton et al. (1988). Clifton et al. (1988) determined the reaction rate from the consumption rate the reactant, NO$_2$, which corresponds to the first reaction step of NO$_2$ with S(IV). Yet, Lee and Schwartz (1983) determined the reaction rate from the production rate of products (their conductivity), which is expected to be much slower than NO$_2$ consumption since formation of products needs more steps. In this study, we determined γ using sulfate production rate, and thus our data are comparable to the study of Lee and Schwartz (1983). Yet, the study of Lee and Schwartz (1983) only covers a pH range of 5-6.4 and has no overlap with the pH (~3) in our study, therefore uptake coefficients from both studies are not directly comparable. Nevertheless, the reaction rate of 1.4×10$^5$ mol$^{-1}$ L$^{-1}$ s$^{-1}$ at pH 5 corresponds to the uptake coefficient of 4.3×10$^{-7}$, which is around one order of magnitude higher than the uptake coefficient in our study determined at pH ~3 for the droplet. The difference may be due to the different pH between these two studies, the different mechanisms between the multiphase reaction on particles and bulk aqueous reaction, and the different concentrations of each S(IV) species since the different species may have different reactivity with NO$_2$. The reaction rate of S(IV) has been found to decrease with decreasing pH and the reactivity of sulfite with NO$_2$ seems to be higher than bisulfite (Lee and Schwartz, 1983; Clifton et al., 1988; Takeuchi et al., 1977). In addition, the ionic strength in the droplet of this study (15-55 mol Kg$^{-1}$) was much higher than that in the bulk solution in previous studies (on the order of 10$^{-6}$-10$^{-1}$ mol Kg$^{-1}$), which may also influence the reaction rate."

*2. Based on Equation (5), the authors concluded that the precipitation-induced reduction of sulfate will promote the oxidation of SO$_2$ by NO$_2$ (reaction 2). I don't know if it is correct to use Eq. (5) in this way. Because Equation (5) is valid for reversible reactions and removing/adding products of non-reversible reactions will not change the reaction rate much.*

**Response:**

Using the change of Gibbs energy to express the spontaneity of a reaction is applicable to all reactions. Moreover, in theory, all chemical reactions are reversible, to some extent (Keeler and Wothers, 2008; de Nevers, 2012). According to the different results between the reaction on Ca(NO$_3$)$_2$ droplet and the reaction on NaNO$_3$ and NH$_4$NO$_3$ droplet, there might be a backward reaction of SO$_2$ oxidation which consumed sulfate, although the detailed mechanism of the backward reaction is unknown at the moment. Therefore, we used Equation (5) to explain the difference between the reaction on Ca(NO$_3$)$_2$ droplet and the reaction on NaNO$_3$ and NH$_4$NO$_3$ droplet.

In the revised manuscript, we have modified the discussion:

"$\Delta_r G$ increases with increasing sulfate concentration. According to the different results between the reaction on $Ca(NO_3)_2$ droplet and the reaction on $NaNO_3$ and $NH_4NO_3$ droplet, there might be a backward reaction of $SO_2$ oxidation which consumed sulfate, although the detailed mechanism of the backward reaction is unknown at the moment. For $NaNO_3$ and $NH_4NO_3$ droplet, once sulfate concentration reached certain level, the reaction may stop due to the increase of $\Delta_r G$. For $Ca(NO_3)_2$ droplet, the precipitation of $CaSO_4$ can substantially decrease the activity of $SO_4^{2-}$, and thus decrease $\Delta_r G$ and promote the oxidation of $SO_2$ and sulfate formation."

*Other comments:*

*Page 5 line 133, half sentene?*

**Response:**

Accepted. In the revised manuscript, we have fixed this sentence. Now it reads:

"In order to minimize the influence variations of incident laser on Raman intensity, these seven particles were measured before each experiment..."

*Page 6 line 187, I would suggest to briefly describe the mechanism of $Ca(NO_3)_2$ formation.*

**Response:**

Accepted. In the revised, we have briefly described the reaction mechanism of $Ca(NO_3)_2$ formation and provided a brief mechanism of the reaction of $CaCO_3$ with $NO_2$. Now it reads:

"$Ca(NO_3)_2$ has been observed in the reaction of $CaCO_3$ with $NO_2$ in previous studies (Li et al., 2010; Tan et al., 2016). The formation of $Ca(NO_3)_2$ started with the reaction of $NO_2$ with adsorbed or liquid water, forming $HNO_3$ and $HNO_2$. Then $HNO_3$ reacted with $CaCO_3$ forming $Ca(NO_3)_2$ as well as $CO_2$, which was released to the gas phase. The reaction equations are as follows:

$$NO_2(g) \leftrightarrow NO_2(aq) \tag{R1}$$

$$2NO_2(aq) + H_2O(aq) \longrightarrow HNO_3(aq) + HNO_2(aq) \tag{R2}$$

$$HNO_3(aq) \longrightarrow H^+(aq) + NO_3^-(aq) \tag{R3}$$

$$CaCO_3(s) + H^+(aq) \longrightarrow Ca^+(aq) + HCO_3^-(aq) \tag{R4}$$

$$HCO_3^-(aq) + H^+(aq) \longrightarrow H_2O(aq) + CO_2(g) \tag{R5}$$"

*Will the present of $SO_2$ influence the uptake of $NO_2$?*

**Response:**

In principle, the presence of $SO_2$ should enhance $NO_2$ uptake due to its reaction with $NO_2$. However, as we observed in our study, the reactive uptake of $NO_2$ on $CaCO_3$ particles was much faster than the reaction of $NO_2$ with $SO_2$ and sulfate as the reaction product of $SO_2$ was essentially only observed after $CaCO_3$ was completely converted to $Ca(NO_3)_2$ droplet by $NO_2$. Therefore, the influence of $SO_2$ on $NO_2$ uptake was not significant.

In the revised manuscript, we have added one sentence to discuss this point.

"The much faster $Ca(NO_3)_2$ formation due to the $NO_2$ uptake on $CaCO_3$ particle compared with the reaction of $SO_2$ with $NO_2$ and sulfate appearing only after the complete conversion of $CaCO_3$ indicate that the reaction of $SO_2$ with $NO_2$ does not contributed significantly to $NO_2$ uptake."

*Fig. 4, no data for nitrate and carbonate after 120 min, why?*

**Response:**

By 118 min, $CaCO_3$ was completely converted to $Ca(NO_3)_2$. Carbonate had decreased to zero and nitrate had reached a plateau. Therefore no further data of carbonate and nitrate were shown. In the revised manuscript, we have explained this in the caption of Fig. 4.

$$\frac{1}{\gamma} = \frac{1}{\Gamma_{diff}} + \frac{1}{\alpha} + \frac{1}{\Gamma_{sat} + \Gamma_{rxn}} \tag{1}$$

where $\Gamma_{diff}$ is the transport coefficient in the gas phase, $1/\Gamma_{diff}$ is the resistance due to the diffusion in the gas phase. Similarly, $1/\Gamma_{sat}$ and $1/\Gamma_{rxn}$ are the resistance due to liquid phase saturation and liquid phase reaction, respectively. $\alpha$ is the mass accommodation coefficient of $SO_2$.

$1/\Gamma_{diff}$ can be derived using the following equation:

$$\frac{1}{\Gamma_{diff}} = \frac{0.75 + 0.238 Kn}{Kn(1 + Kn)} . \tag{2}$$

where Kn is the Knudsen number. Knudsen number is defined as

$$Kn = \frac{\lambda}{a} , \tag{3}$$

where $\lambda$ is the mean free path of molecule in the gas phase and a is the radius of the particle.

$\lambda$ can be derived from

$$\lambda = \frac{3 D_g}{c}, \tag{4}$$

where $D_g$ is the diffusion coefficient in the gas phase and c is the mean molecular velocity.

c is derived from

$$c = \sqrt{\frac{8RT}{\pi M}} \tag{5}$$

where R is the gas constant, T is temperature, and M is the molecular mass of $SO_2$.

$1/\Gamma_{sat}$ can be derived from

$$\frac{1}{\Gamma_{sat}} = \frac{c}{4HRT} \sqrt{\frac{t\pi}{D_l}} , \tag{6}$$

where H is the Henry constant of $SO_2$, t is time, and $D_l$ is the diffusion coefficient of $SO_2$ in the liquid phase.

$1/\Gamma_{rxn}$ can be derived from

$$\frac{1}{\Gamma_{rxn}} = \frac{c}{4HRT} \sqrt{\frac{1}{k_{rxn} D_l}} , \tag{7}$$

where $k_{rxn}$ is the first order rate constant of the reaction in the liquid phase.

$$k_{rxn} = k[NO_2(aq)], \tag{8}$$

where k is the second order rate constant of the reaction of S(IV) with $NO_2$ and $[NO_2(aq)]$ is the $NO_2$ concentration in the liquid phase.

$$[NO_2(aq)] = H_{NO2} P_{NO2}, \tag{9}$$

where $H_{NO2}$ is the Henry constant of $NO_2$ and $P_{NO2}$ is the concentration of $NO_2$ in the gas phase.

**Table S1** Constants used for deriving uptake coefficients from reaction rates.

| Parameter | Value | Reference |
|---|---|---|
| $D_g$ ($m^2$ $s^{-1}$) | $10^{-5}$ | - |
| $a$ (m) | $8.3 \times 10^{-6}$ | - |
| $R$ ($J$ $mol^{-1}$ $K^{-1}$) | 8.314 | - |
| $T$ (K) | 298 | - |
| $M_{SO2}$ ($Kg$ $mol^{-1}$) | $6.4 \times 10^{-2}$ | - |
| $\alpha$ | 0.35 | Davidovits et al. (2006) |
| $D_l$ ($m^2$ $s^{-1}$) | $8.3 \times 10^{-12a}$ | Mahiuddin and Ismail (1983) |
| $H_{SO2}$ ($mol$ $L^{-1}$ $atm^{-1}$) | 1.23 | Seinfeld and Pandis (2006) |
| $H_{NO2}$ ($mol$ $L^{-1}$ $atm^{-1}$) | $1 \times 10^{-2}$ | Seinfeld and Pandis (2006) |

[a]The aqueous phase diffusion coefficient was derived from the viscosity of $Ca(NO_3)_2$ solution providing that diffusion coefficient is inversely proportional to viscosity according to the Stokes−Einstein equation (Bones et al.,

2012) and assuming that the diffusion coefficient in water is $10^{-9}$ $m^2$ $s^{-1}$.

[Figure]

Figure S1. Calibration curve for sulfate showing the peak area of sulfate at 1016 cm$^{-1}$ in Raman spectra versus the amount of CaSO$_4$.

[Figure]

Figure S2. Raman mapping analysis of a CaCO$_3$ particle during the reaction with NO$_2$ **(**75 ppm) and SO$_2$ (75 ppm)

at 72% RH at the reaction time of 0, 8, 26, 40, 97, and 1053 min. Blue, red, and green indicate the Raman peak intensity of carbonate, nitrate, and sulfate at 1087, 1050, and 1013 cm$^{-1}$, respectively.

---

## Referee Report (RR1)

**Review ACP-2017-610**

The authors answered the comments of both reviewers correctly, with some exceptions listed below.

**Reviewer 1.**

**Comment 8.** [lines 232-236] The concentrations of $HSO_3^-$, $SO_3^{2-}$ and $H_2SO_3$ in equilibrium with gaseous $SO_2$ (75 ppm ?, specify the constants used, please, not the reference alone) may not be the actual concentrations in the droplets due to chemical reactions involved. Good answer needs some modeling.

**Comment 9.** [lines 247-263] Unanimously with Reviewer 1, I think that $SO_2$ should react with $NO_2$ not only in $Ca(NO_3)_2$ droplets but also in $NaNO_3$ and $NH_4NO_3$ ones. The authors claimed any sulfate produced was below the limit of detection in the latter systems. Their Raman spectra included only $SO_4^{2-}$ ions from anhydrite ($CaSO_4$). No crystals were observed in the absence of calcium ions, clearly because the solubility of sodium or ammonium sulfate in water is much higher than that of $CaSO_4$.

Thus, the experiments presented in the manuscript are inadequate to conclude on the reactive uptake of $SO_2$ and $NO_2$ in $NaNO_3$ and $NH_4NO_3$ droplets and formation of sulfate therein. We just do not know how much sulfate was formed in these experiments. N.B., it would be interesting to know all the limits of detection involved as there were other species undetected in the droplets, e.g. sulfite ion, $SO_3^{2-}$ and $CaSO_3$ [lines 239-244].

The comment on using eqn (5) and Gibbs free energy of the reaction is provided just below (Reviewer 2).

**Reviewer 2.**

**Major concern 2.** The authors are right that eqn (5) is valid for all reactions (an equilibrium takes place when $\Delta G = 0$). However, it is difficult if possible to use eqn (5) to prove that precipitation of $CaSO_4$ crystals promotes sulfate formation by reducing the concentration of sulfate in the reaction environment. The authors claimed [lines 257-263] that some backward reaction yet unknown can bring back $SO_2$ from sulfate so that increased concentration of sulfate ions can slow down the formation of sulfate ions while precipitation keeps the concentration of sulfate ions low. In my opinion, under conditions of their experiments, such backward reaction is just impossible. The sulfate ions could slow down the formation of sulfate only by crowding the reaction environment and reducing the encounter probability for the reactants. Maybe, by decreasing the uptake of reagents from the gas phase as well. Precipitation to $CaSO_4$ crystals removed the nuisance, naturally.

[line 253] Replace "Reaction (R2)" with "Reactions (R6) and (R7)".

**Comment:** *Will the presence of $SO_2$ influence the uptake of $NO_2$?*

The authors somehow missed their response [lines 201-2014] that the influence of $SO_2$ on $NO_2$ uptake was not significant. Following their line, it was, probably, not significant when calcium carbonate was

present. When there was no carbonate anymore, the reaction of $SO_2$ with $NO_2$ plausibly increased the uptake of both gases by $Ca(NO_3)_2$ solution.

**My own comments:**

(1) The role or fate of $NO_2^-$/HONO (reactions R2, 6 and 7) has not been commented in the manuscript, even though this product is quite reactive.

(2) The uncertainties of uptake coefficients in Table 2 should be explained.

(3) [Lines 226-229] R6 and R7 are overall stoichiometric reactions but not a reaction mechanism.

**The question whether the studies of $SO_2$ oxidation by $NO_2$ (acp-2017-610) and on $SO_2$ oxidation by $O_2$ (acp-2017-900) should be combined together or not.**

I support Reviewer 1 in her/his opinion that the two manuscripts should be combined in one well-structured work. It is a natural desire of a reader, when exploring the first paper, to compare immediately and easily the experiments presented to those including oxygen, a common atmospheric reactant. The authors had right to divide their work as they wished. They stated their reasons for doing so but they did not convince me. There are many common parts in both manuscripts while discussion and conclusions are complemental. However, I do not hesitate to remind that researchers are evaluated basing on the number of papers and citations, and the Hirsch index. Especially when they apply for grants or positions. Therefore, we are often tempted to produce 2 or 3 papers rather than a single well-structured publication.

---

## Author Response (AR3)

**Response to the editor**

We thank the editor for the time and the helpful comments on our manuscript. We have addressed all the comments. According to these comments, we have revised our manuscript. As follows, please find our one-by-one responses to these comments. The original comments are shown in italics and the revised texts are highlighted.

*Comments to the Author:*

*dear authors*

*I acknowledge that the reaction studied in this manuscript, $NO_2$ + $SO_2$, in absence of $O_2$, is important enough to merit publication as a separate manuscript. The formation of sulfate from S(IV) in aerosol particles especially under heavy pollution conditions has been suggested to operate with $NO_2$ being the oxidant. Even though this study has not covered the atmospherically relevant $NO_2$ partial pressure range and also has not looked at the effect of pH or especially high pH values for which the reaction $NO_2$ + $SO_2$ has been suggested to contribute most to the formation of sulfate, it provides valuable insight into the mechanism of this reaction at low pH (relevant for most atmospheric conditions). It is important to document the low reaction rates resulting from this reaction.*

**Response:**

We thank the editor for the supportive remarks on the scientific significance of our study.

*To make sure that these aspects are well documented and explained I am asking for a few minor additional revisions in the relevant sections of the manuscript:*

*abstract, line 17: please also mention that it is not important also in comparison to the other aqueous phase pathways, such as reaction with $O_2$, w/o transition metals, or $H_2O_2$.*

**Response:**

Accepted. In the revised manuscript, we have revised the sentence as follows:

"We estimate that the direct multiphase oxidation of $SO_2$ by $NO_2$ is not an important source of sulfate in the ambient atmosphere compared with the $SO_2$ oxidation by OH in the gas phase and is not as important as other aqueous phase pathways, such as the reactions of $SO_2$ with $H_2O_2$, $O_3$, and $O_2$, with or without transition metals."

*line 38: replace 'liquid water' by aqueous solution. Strictly speaking, 'liquid water' refers to pure water only. please check the usage of the term 'liquid water' also in the remainder of the manuscript.*

**Response:**

Accepted. In the revised manuscript, we have replaced "liquid water" with "aqueous solution" here and checked its usage throughout the manuscript. In some parts, we have replaced it with "water in aqueous solution".

*line 63: ionic strength (not ion strength)*

**Response:**

Accepted. In the revised manuscript, we have corrected it.

*line 67 and 78: mention the contribution of this reaction to sulfate formation in the Cheng /Wang / Xue studies in comparison to that involving $O_2$ (if they looked at it). For the Cheng et al. reference: explicitly mention that they considered this reaction (not with $O_2$) being the most important pathway.*

**Response:**

Accepted.

The studies of Cheng et al., 2016/Wang et al., 2016/Xue et al., 2016 all focus on the contribution of the direct oxidation of $SO_2$ by $NO_2$ to sulfate formation. The reaction involving $O_2$ was only investigated in Wang et al., (2016) and was found to be negligible in sulfate formation. In the revised manuscript, we have mentioned this. We have also explicitly mentioned that the study of Cheng et al. (2016) considered the direct oxidation of $SO_2$ by $NO_2$ to be the most important sulfate formation pathway. The revised text is as follows:

"In this study, we present the finding that the multiphase reaction of $SO_2$ directly with $NO_2$ is not an important source of sulfate in the atmosphere, in the absence of other oxidants such as $O_2$. The direct oxidation of $SO_2$ by $NO_2$ pathway was proposed in a number of recent studies to be potentially important for sulfate formation (Cheng et al., 2016; Wang et al., 2016; Xue et al., 2016). For example, Cheng et al. (2016) considered the direct oxidation of $SO_2$ by $NO_2$ to be the most important pathway to explain the missing sulfate source during the haze events in Beijing. Wang et al. (2016) also proposed that the direct oxidation of $SO_2$ by $NO_2$ is key to efficient sulfate formation in the presence of high relative humidity and $NH_3$ and showed that in their laboratory study sulfate formation is mainly contributed by the direct oxidation by $NO_2$ and the role of $O_2$ is negligible."

*line 114: maybe explain why the majority of experiments were not performed with $Ca(NO_3)_2$ directly. I don't think that reaction of $CaCO_3$ with $NO_2$ is the main source of $Ca(NO_3)_2$ in the atmosphere. I guess $HNO_3$ is doing this mostly.*

**Response:**

Accepted.

Most of the experiments in this study were conducted using $CaCO_3$ rather than directly using $Ca(NO_3)_2$ because $CaCO_3$ is an important component of mineral aerosols, especially in China as mentioned in the introduction and often used as a surrogate of mineral aerosols. Moreover, using $CaCO_3$ particles can better simulate the reaction on internally-mixed $CaCO_3$(solid)-$Ca(NO_3)_2$(aqueous) particles, which is widely observed in the ambient atmosphere and laboratory (Laskin et al., 2005; Zhang et al., 2003; Li and Shao, 2009; Sullivan et al., 2007; Li et al., 2010; Liu et al., 2008), and is formed via the reaction of $CaCO_3$ with acidic gases such as $HNO_3$ and $NO_2$ due to its alkalinity.

$Ca(NO_3)_2$ is usually believed to be mainly formed by the reaction of $CaCO_3$ with $HNO_3$. The reaction of $CaCO_3$ with $NO_2$ also contributes to the formation of $Ca(NO_3)_2$, especially at high $NO_2$ and at high RH (Li et al., 2010). Regardless of the source of $Ca(NO_3)_2$, $Ca(NO_3)_2$ droplet or $Ca(NO_3)_2$ aqueous layer on $CaCO_3$ particles can provide a site for the multiphase oxidation of $SO_2$ by $NO_2$.

In the revised manuscript, we have revised the text as follows.

"Most experiments were conducted using $CaCO_3$ particles rather than directly using $Ca(NO_3)_2$ particles. $CaCO_3$ was selected because it is an important component of mineral aerosols, especially in China as mentioned in the introduction and often used as a surrogate of mineral aerosols. Moreover, using $CaCO_3$ particles can better simulate the reaction on internally-mixed $CaCO_3$(solid)-$Ca(NO_3)_2$(aqueous) particles, which is widely observed in the ambient atmosphere and laboratory (Laskin et al., 2005; Zhang et al., 2003; Li and Shao, 2009; Sullivan et al., 2007; Li et al., 2010; Liu et al., 2008), and is formed via the reaction of $CaCO_3$ with acidic gases such as $HNO_3$ and $NO_2$ due to its alkalinity."

*line 160: of $H_2O$ in aqueous solution.*

**Response:**

Accepted. In the revised manuscript, we have changed "liquid water" to "water in aqueous solution".

*line 364: add a caveat here that this study did not look at the pH dependence, and especially not at high pH, for which recent studies have claimed this reaction to be important. Also please mention that more likely other oxidation pathways, such as with $H_2O_2$, $O_3$ and $O_2$ as oxidants, and w/o transition metals, must be more important.*

**Response:**

Accepted.

In the revised manuscript, we have added the following discussion as the editor suggested.

"It is worth mentioning that this study did not investigate the dependence of the reactive uptake coefficient due to the direct oxidation of $SO_2$ by $NO_2$ on pH, especially not under high pH conditions, for which recent studies have claimed this reaction to be important (Cheng et al., 2016; Wang et al., 2016). Because of the important role of multiphase/heterogeneous reactions in $SO_2$ oxidation found in the atmosphere and the low reaction rate of the direct multiphase oxidation of $SO_2$ by $NO_2$, it is more likely that the aqueous reactions of $SO_2$ with other oxidants, such as the reactions with $H_2O_2$, $O_3$, and $O_2$, with or without transition metals, could be important pathways for sulfate formation in the atmosphere."

*lines around 370: I think the statements about the mixing state of $CaSO_4$ with nitrate are ok, but this is independent of the oxidation mechanism; but the source of nitrate is more likely $HNO_3$ and $N_2O_5$, and rather not $NO_2$ under atmospheric conditions.*

**Response:**

Accepted.

We agree that the formation of internally-mixed $CaSO_4$/$Ca(NO_3)_2$ particles is independent of the oxidation mechanism of $SO_2$ once $Ca(NO_3)_2$ aqueous layer is formed via the reaction with either $HNO_3$, $N_2O_5$, or $NO_2$, although the reaction of $SO_2$ with $NO_2$ is shown to contribute to the formation of these internally-mixed particles in this study. In the revised manuscript, we decide to delete this paragraph.

**References**

[revised manuscript text omitted]

**S1. Deriving the reactive uptake coefficient from aqueous reaction rate constant**

The reaction rate constant in the literature was used to derive the reactive uptake coefficient to particles under the same conditions as in this study using the method in Davidovits et al. (2006).

$$\frac{1}{\gamma} = \frac{1}{\Gamma_{diff}} + \frac{1}{\alpha} + \frac{1}{\Gamma_{sat} + \Gamma_{rxn}} \tag{S1}$$

where $\Gamma_{diff}$ is the transport coefficient in the gas phase, $1/\Gamma_{diff}$ is the resistance due to the diffusion in the gas phase. Similarly, $1/\Gamma_{sat}$ and $1/\Gamma_{rxn}$ are the resistance due to liquid phase saturation and liquid phase reaction, respectively. $\alpha$ is the mass accommodation coefficient of $SO_2$.

$1/\Gamma_{diff}$ can be derived using the following equation:

$$\frac{1}{\Gamma_{diff}} = \frac{0.75 + 0.238Kn}{Kn(1+Kn)} \tag{S2}$$

where Kn is the Knudsen number. Knudsen number is defined as

$$Kn = \frac{\lambda}{a} \tag{S3}$$

where $\lambda$ is the mean free path of molecule in the gas phase and a is the radius of the particle.

$\lambda$ can be derived from

$$\lambda = \frac{3D_g}{c} \tag{S4}$$

where $D_g$ is the diffusion coefficient in the gas phase and c is the mean molecular velocity.

c is derived from

$$c = \sqrt{\frac{8RT}{\pi M}} \tag{S5}$$

where R is the gas constant, T is temperature, and M is the molecular mass of $SO_2$.

$1/\Gamma_{sat}$ can be derived from

$$\frac{1}{\Gamma_{sat}} = \frac{c}{4HRT}\sqrt{\frac{t\pi}{D_l}} \tag{S6}$$

where H is the Henry constant of $SO_2$, t is time, and $D_l$ is the diffusion coefficient of $SO_2$ in the liquid phase.

$1/\Gamma_{rxn}$ can be derived from

$$\frac{1}{\Gamma_{rxn}} = \frac{c}{4HRT}\sqrt{\frac{1}{k_{rxn}D_l}} \tag{S7}$$

where $k_{rxn}$ is the first order rate constant of the reaction in the liquid phase.

$$k_{rxn} = k[NO_2(aq)] \tag{S8}$$

where k is the second order rate constant of the reaction of S(IV) with $NO_2$ and $[NO_2(aq)]$ is the $NO_2$ concentration in the liquid phase.

$$[NO_2(aq)] = H_{NO2}P_{NO2} \tag{S9}$$

where $H_{NO2}$ is the Henry constant of $NO_2$ and $P_{NO2}$ is the concentration of $NO_2$ in the gas phase.

**S2. Characteristic time for aqueous reaction and gas-particle equilibrium**

The characteristic time to achieve the equilibrium in the gas-particle interface and for aqueous reaction of $SO_2$ with $NO_2$ were derived using the method in Seinfeld and Pandis (2006).

$$\tau_p \cong \frac{aH^* \sqrt{2\pi MRT}}{3\alpha} \tag{S10}$$

$$\tau_{ra} = \frac{1}{k_{rxn}} \tag{S11}$$

a is the radius of the particle, $H^*$ is the effective Henry constant, M is the molecular weight, R is the gas constant, T is temperature, $\alpha$ is the mass accommodation coefficient of $SO_2$. $k_{rxn}$ is the first-order rate constant of the reaction in the liquid phase (see Equation S8). The values of the constants are shown in Table S1.

The characteristic time to achieve the equilibrium in the gas-particle interface is around $4\times10^{-5}$ s. The characteristic time for aqueous reaction is 0.5 and 0.08 s using the reaction rate constant of $2\times10^{6}$ $mol^{-1}$ L $s^{-1}$ (Lee and Schwartz, 1983) and $1.24\times10^{7}$ $mol^{-1}$ L $s^{-1}$ (Clifton et al., 1988), respectively.

**Table S1** Constants used for deriving uptake coefficients from reaction rates and deriving characteristic time

| Parameter | Value | Reference |
|---|---|---|
| $D_g$ (m$^2$ s$^{-1}$) | $10^{-5}$ | - |
| a (m) | $8.3 \times 10^{-6}$ | - |
| R (J mol$^{-1}$ K$^{-1}$) | 8.314 | - |
| T (K) | 298 | - |
| $M_{SO2}$ (Kg mol$^{-1}$) | $6.4 \times 10^{-2}$ | - |
| α | 0.35 | Davidovits et al. (2006) |
| $D_l$ (m$^2$ s$^{-1}$) | $8.3 \times 10^{-12a}$ | Mahiuddin and Ismail (1983) |
| $H_{SO2}$ (mol L$^{-1}$ atm$^{-1}$) | 1.23 | Seinfeld and Pandis (2006) |
| $H_{NO2}$ (mol L$^{-1}$ atm$^{-1}$) | $1 \times 10^{-2}$ | Seinfeld and Pandis (2006) |
| k (mol$^{-1}$ L s$^{-1}$) | $2 \times 10^6$
 $1.24 \times 10^7$ | Lee and Schwartz (1983);
 Clifton et al. (1988) |

[a]The aqueous phase diffusion coefficient was derived from the viscosity of $Ca(NO_3)_2$ solution providing that
diffusion coefficient is inversely proportional to viscosity according to the Stokes−Einstein equation (Bones et al.,
2012) and assuming that the diffusion coefficient in water is $10^{-9}$ m$^2$ s$^{-1}$.

[Figure]

Figure S1. Calibration curve for sulfate showing the peak area of sulfate at 1016 cm$^{-1}$ in Raman spectra versus the amount of CaSO$_4$.

[Figure]

Figure S2. Raman mapping analysis of a CaCO$_3$ particle during the reaction with NO$_2$ **(**75 ppm) and SO$_2$ (75 ppm)

at 72% RH at the reaction time of 0, 8, 26, 40, 97, and 1053 min. Blue, red, and green indicate the Raman peak intensity of carbonate, nitrate, and sulfate at 1087, 1050, and 1013 cm$^{-1}$, respectively.